# Molecular shifts in limb identity underlie development of feathered feet in two domestic avian species

**Eric T Domyan[1†], Zev Kronenberg[2‡], Carlos R Infante[3§], Anna I Vickrey[1], Sydney A Stringham[1], Rebecca Bruders[1], Michael W Guernsey[1¶], Sungdae Park[3], Jason Payne[4], Robert B Beckstead[4], Gabrielle Kardon[2], Douglas B Menke[3], Mark Yandell[2,5], Michael D Shapiro[1]\***

[1]Department of Biology, University of Utah, Salt Lake City, United States; [2]Department of Human Genetics, University of Utah, Salt Lake City, United States; [3]Department of Genetics, University of Georgia, Athens, United States; [4]Poultry Science Department, University of Georgia, Athens, United States; [5]Utah Center for Genetic Discovery, University of Utah, Salt Lake City, United States

**\*For correspondence:** shapiro@biology.utah.edu

**Present address:** [†]Department of Biology, Utah Valley University, Orem, United States; [‡]Department of Genome Sciences, University of Washington, Seattle, United States; [§]Department of Molecular and Cellular Biology, University of Arizona, Tucson, United States; [¶]Department of Genetics, Stanford University, Stanford, United States

**Competing interests:** The authors declare that no competing interests exist.

**Abstract** Birds display remarkable diversity in the distribution and morphology of scales and feathers on their feet, yet the genetic and developmental mechanisms governing this diversity remain unknown. Domestic pigeons have striking variation in foot feathering within a single species, providing a tractable model to investigate the molecular basis of skin appendage differences. We found that feathered feet in pigeons result from a partial transformation from hindlimb to forelimb identity mediated by *cis*-regulatory changes in the genes encoding the hindlimb-specific transcription factor Pitx1 and forelimb-specific transcription factor Tbx5. We also found that ectopic expression of *Tbx5* is associated with foot feathers in chickens, suggesting similar molecular pathways underlie phenotypic convergence between these two species. These results show how changes in expression of regional patterning genes can generate localized changes in organ fate and morphology, and provide viable molecular mechanisms for diversity in hindlimb scale and feather distribution.

## Introduction

In birds, the genetic and developmental mechanisms that control the decision between scale and feather development remain poorly understood. Most birds possess scales on the foot (tarsometa-tarsus and toes) and feathers elsewhere. Exceptions to this pattern can provide insights into the evolutionary and developmental basis of skin appendage diversity. Some raptors and boreal birds evolved foot feathers instead of scales ('ptilopody'; *Danforth, 1919*; *Lucas and Stettenheim, 1972*), but the lack of appendage variation *within* these species precludes their use as genetic models. Likewise, paravians (birds and their close non-avian theropod dinosaur relatives) and other dinosaurs evolved diverse feather coverings on their legs and feet that sometimes resemble flight-like feathers (*Xu et al., 2003*; *Hu et al., 2009*; *Turner et al., 2012*; *Zheng et al., 2013*; *Foth et al., 2014*; *Godefroit et al., 2014*), but the absence of living specimens preclude mechanistic molecular studies.

In contrast, domestic pigeons (*Columba livia*) exhibit stunning variation within a single extant species (*Shapiro and Domyan, 2013*). Most breeds have feet covered by scaled epidermis (wild-type), but scales are replaced by small feathers in *grouse (gr)* mutants, and by larger feather 'muffs' in birds that also carry mutant alleles at the *Slipper (Sl)* locus (*Doncaster, 1912*; *Wexelsen, 1934*;

**eLife digest** Animals ranging from fish to birds display dramatic diversity within and among species; yet remarkably little is known about the genetic and developmental mechanisms that underlie this variation. In birds and their extinct dinosaur relatives, the distribution of scales and feathers on the feet is a highly variable trait.

Different breeds of domestic pigeon all belong to the same species but have feet that can be feathery or scaly to different extents. Classical genetics experiments suggested that only a few genes are involved in the transition from scaled to feathered skin on the feet of pigeons. However, the molecular basis for this transition was unknown.

Domyan et al. set out to identify the genes involved in the transition from scaled to feathered feet by mating different breeds of pigeon in the laboratory and then sequencing the birds' DNA. They also surveyed the entire DNA sequences of many additional pigeon breeds with and without feathered feet. This combined approach showed that two regions of the pigeon genome have a profound effect on the number and size of feathers on the feet of domestic pigeons. These regions contain genes that are known to play key roles in controlling the growth of a limb and whether it develops into a leg or a wing. In developing pigeon embryos, Domyan et al. found that a gene called *Pitx1,* which is typically considered a hindlimb gene, is expressed at lower levels in the developing legs of breeds with feathered feet than in a breed with scaled feet. The experiments also showed that *Tbx5,* a gene that is expressed in the forelimbs of many animals, is expressed abnormally in the embryonic hindlimbs of breeds of pigeon and chicken with feathery feet.

Together, these findings suggest that the hindlimbs of domestic birds with feathery feet are more like wings at the molecular level, which results in them being covered in feathers rather than scales. Future work will now aim to discover the specific DNA sequences that alter the expression of *Pitx1* and *Tbx5* in feather-footed breeds, and whether the same genes control the foot feathers of other species of birds.

*Hollander, 1937*; *Levi, 1986*) (*Figure 1A*). In muffed breeds, scutellate scales are generally absent or poorly developed on the feathered epidermis covering the tarsometatarsus and toes, and feathers are surrounded by soft integument. The molecular identities of both *gr* and *Sl* are unknown, and additional loci probably control quantitative variation in the muff phenotype. Because both scale-footed and feather-footed pigeon breeds belong to the same species, we can use traditional genetic crosses and whole-genome resequencing to map the genes that control this striking variation (*Shapiro et al., 2013*; *Domyan et al., 2014*). Therefore, we can study diversity of the magnitude usually observed *among* different species without the roadblock of hybrid incompatibility that often eliminates the possibility of genetic mapping studies.

During development in vertebrates, skin appendages form through interactions between the ectoderm-derived epidermis and the mesoderm-derived dermis, and signals from the dermis determine epidermal appendage fate (*Dhouailly, 2009*; *Hughes et al., 2011*). Previous analyses of mutants and gene misexpression in chickens suggest candidates for feathered feet in the *Hedgehog* (*Harris et al., 2002*), *BMP* (*Zou and Niswander, 1996*; *Harris et al., 2002*; *2004*), *Delta-Notch* (*Crowe et al., 1998*), and *Wnt* (*Chang et al., 2004*) pathways. Our study of genetic variation and embryonic development in pigeons, however, reveals a surprisingly different mechanism with broad implications for limb identity and patterning.

## Results

### Two loci of major effect control foot feathering in pigeons

To identify chromosome regions that contribute to feathered feet, we generated an $F_2$ intercross between muffed (Pomeranian pouter) and scaled (Scandaroon) breeds (*Figure 1—figure supplement 1*). $F_1$ hybrids displayed intermediate foot feathering, demonstrating a semi-dominant inheritance pattern. Scaled, muffed, and intermediate phenotypes were recovered in the $F_2$ population, confirming that a small number of major-effect loci contribute to this trait. Among $F_2$ offspring, digit

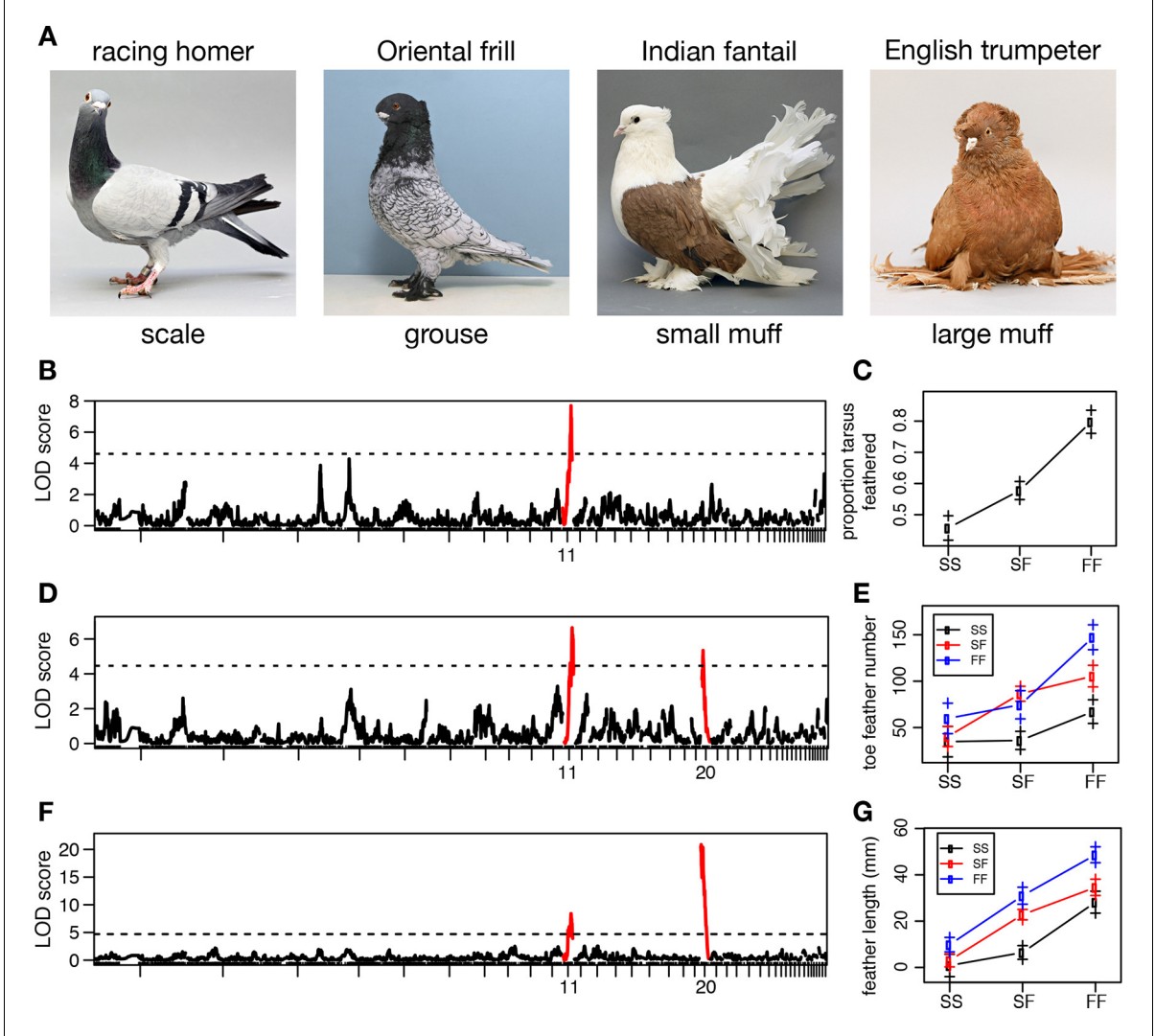

**Figure 1.** Two QTL differentiate scale- and feather-footed domestic pigeons. (A) Common phenotypes of domestic rock pigeon, in order of increasing foot feathering (left to right): scaled, groused, small- and large-muffed. (B-F) QTL scans and effect plots: proportion of tarsus feathered (B,C), number of toe feathers (D,E), and length of toe feathers (F,G). Mean phenotypes ± S.E. are plotted in (C,E,G). For (E) and (G), genotypes at the QTL with the higher LOD score are on the x-axis, and genotypes at the other QTL are inset. See *Tables 1* and *2* for detailed QTL statistics. S, allele from scale-footed grandparent; F, allele from feather-footed grandparent.

The following figure supplement is available for figure 1:

**Figure supplement 1.** Foot-feathering phenotypes of genetic cross.

3 bore the largest and greatest number of feathers (digit 1, 7.04 ± 5.08 feathers; digit 2, 7.64 ± 7.54; digit 3, 15.46 ± 10.43; digit 4, 6.13 ± 5.70). Using quantitative trait locus (QTL) mapping with 130 $F_2$ offspring genotyped at 3803 polymorphic markers (*Broman et al., 2003*), we identified two linkage groups (LG11 and LG20) that had significant effects on three different aspects of foot feathering ($\log_{10}$ odds ratio (LOD) > 4.6; *Figure 1B–G*, *Table 1*). LG11 had the largest effects on the proportion of the tarsometatarsal epidermis that was transformed from scaled to feathered (LOD = 7.69, percent variance explained (PVE) = 28.4%) and toe feather number (LOD = 6.72, PVE = 21.3%), and a smaller effect on toe feather length (LOD = 8.51, PVE = 15.8%). LG20 had the largest effect on toe feather length (LOD = 20.9, PVE = 52.2%), and a smaller effect on toe feather number (LOD = 5.36, PVE = 16.5%). When toe feather number was analyzed for each digit individually, the same two QTL were identified and had the most pronounced effects on digits 3 and 4 (*Table 2*). In summary, two

**Table 1.** Summary of foot feathering QTL.

| Trait | LG | Loc (cM) | Scaffold | Position | LOD | PVE | Mean ± S.D. | | |
| | | | | | | | SS | SF | FF |
|---|---|---|---|---|---|---|---|---|---|
| Proportion tarsus feathered | 11 | 117 | 79 | 9,205,286 | 7.69 | 28.4 | 0.46 ± 0.04 | 0.58 ± 0.03** | 0.80 ± 0.04*** |
| Number of toe feathers | 11 | 124 | 79 | 12,325,977 | 6.73 | 21.3 | 43.6 ± 8.9 | 67.3 ± 6.5* | 105.3 ± 8.1*** |
| Number of toe feathers | 20 | 15 | 95 | 1,451,127 | 5.36 | 16.5 | 45.3 ± 8.1 | 78.2 ± 6.5** | 100.3 ± 9.9*** |
| Toe feather length (mm) | 20 | 0 | 70 | 136,746 | 20.89 | 52.2 | 5.2 ± 2.3 | 19.9 ± 2.0*** | 37.3 ± 2.5*** |
| Toe feather length (mm) | 11 | 124 | 79 | 12,325,977 | 8.51 | 15.8 | 11.4 ± 3.3 | 18.7 ± 2.4* | 28.5 ± 3.0** |

LG, linkage group; Loc, genetic location of peak LOD score in centimorgans; S, allele from scaled parent; PVE, percent variance explained; F, allele from feathered parent; LOD, log10 odds ratio. (Welch two sample t-test of means compared to SS homozygote; *p≤ 0.05, **p≤0.005, ***p≤0.0005.)

major QTL have marked and separable effects on qualitative and quantitative variation in epidermal appendages.

In parallel to our experimental cross, we performed probabilistic whole-genome scans of allele-frequency differentiation (pFst; see *Kronenberg et al., 2014*) across a genetically and phenotypically diverse panel of breeds by comparing 15 feather-footed birds (4 groused and 11 muffed) to 28 scale-footed birds (*Shapiro et al., 2013*). Using this independent approach *across* breeds, the two most-highly differentiated pFst signals implicate the same genomic regions as the QTL study: genomic scaffold 79 is located on LG11 ($p=4.44 \times 10^{-16}$, genome-wide significance threshold $= 2.11 \times 10^{-9}$), and scaffolds 70 and 95 are adjacent to one another on LG20 ($p=9.81 \times 10^{-13}$) (*Figure 2A*, *Figure 2—figure supplement 1A,B*)

The peak pFst region on scaffold 79 contained a 44-kb deletion (from 6.719–6.763 Mb) that was homozygous in 10, and heterozygous in 2 of the 15 feather-footed birds (*Figure 2B,C*; *Figure 2—figure supplement 2A*). Birds homozygous for the deletion showed elevated levels of haplotype homozygosity relative to scaled birds, a signature of positive selection on this region (*Figure 2B*). This deletion spans an element orthologous to a known human limb enhancer, hs1473 (*Spielmann et al., 2012*), which contains active chromatin marks (*Cotney et al., 2012*) and is bound by the hindlimb-specific transcription factor Pitx1 in the developing mouse hindlimb (*Infante et al., 2013*) (*Figure 2C*). The locus was homozygous for the deletion in 35 of 54 additional feather-footed birds from 21 breeds, but was never homozygous in 96 scale-footed birds from 56 breeds (Chi-square, p<0.0001; *Figure 2—figure supplement 2B*). The 3 feather-footed birds from our whole-genome panel that did not have this deletion (including the male founder of the aforementioned genetic cross implicating this same region) also showed allelic differentiation from scale-footed birds

**Table 2.** Summary of QTL for numbers of feathers on individual toes.

| Digit | LG | Loc (cM) | Scaffold | Position | LOD | PVE | Mean ± S.D. | | |
| | | | | | | | SS | SF | FF |
|---|---|---|---|---|---|---|---|---|---|
| Digit 2, left foot | 11 | 148 | 79 | 11,624,701 | 5.20 | 20.24 | 3.75 ± 1.13 | 6.95 ± 1.12* | 12.19 ± 1.16*** |
| Digit 3, right foot | 20 | 0 | 70 | 136,746 | 7.71 | 24.1 | 9.06 ± 1.70 | 17.35 ± 1.48** | 19.86 ± 1.83*** |
| Digit 3, right foot | 11 | 148 | 79 | 11,624,701 | 6.47 | 19.5 | 9.98 ± 1.60 | 15.60 ± 1.62* | 21.02 ± 1.65*** |
| Digit 3, left foot | 20 | 0 | 70 | 136,746 | 10.79 | 29.84 | 8.66 ± 1.59 | 17.06 ± 1.41** | 21.09 ± 1.86*** |
| Digit 3, left foot | 11 | 148 | 79 | 11,624,701 | 9.48 | 25.52 | 9.37 ± 1.52 | 16.26 ± 1.52** | 21.30 ± 1.57*** |
| Digit 4, right foot | 20 | 32 | 95 | 2,464,788 | 7.33 | 21.36 | 2.44 ± 0.90 | 7.24 ± 0.75** | 10.14 ± 1.10*** |
| Digit 4, right foot | 11 | 118 | 79 | 5,475,474 | 6.78 | 19.5 | 3.17 ± 0.89 | 6.91 ± 0.92* | 9.95 ± 1.00*** |
| Digit 4, left foot | 20 | 30 | 95 | 2,464,788 | 9.37 | 26.83 | 2.44 ± 0.79 | 6.63 ± 0.66** | 9.65 ± 0.96*** |
| Digit 4, left foot | 11 | 148 | 79 | 11,624,701 | 7.58 | 20.83 | 3.75 ± 1.14 | 6.95 ± 1.13* | 12.19 ± 1.17*** |

LG, linkage group; Loc, genetic location of peak LOD score in centimorgans; S, allele from scaled parent; PVE, percent variance explained; F, allele from feathered parent; LOD, log10 odds ratio. (Welch two sample t-test of means compared to SS homozygote; *p≤0.05, **p≤0.005, ***p≤0.0005.)

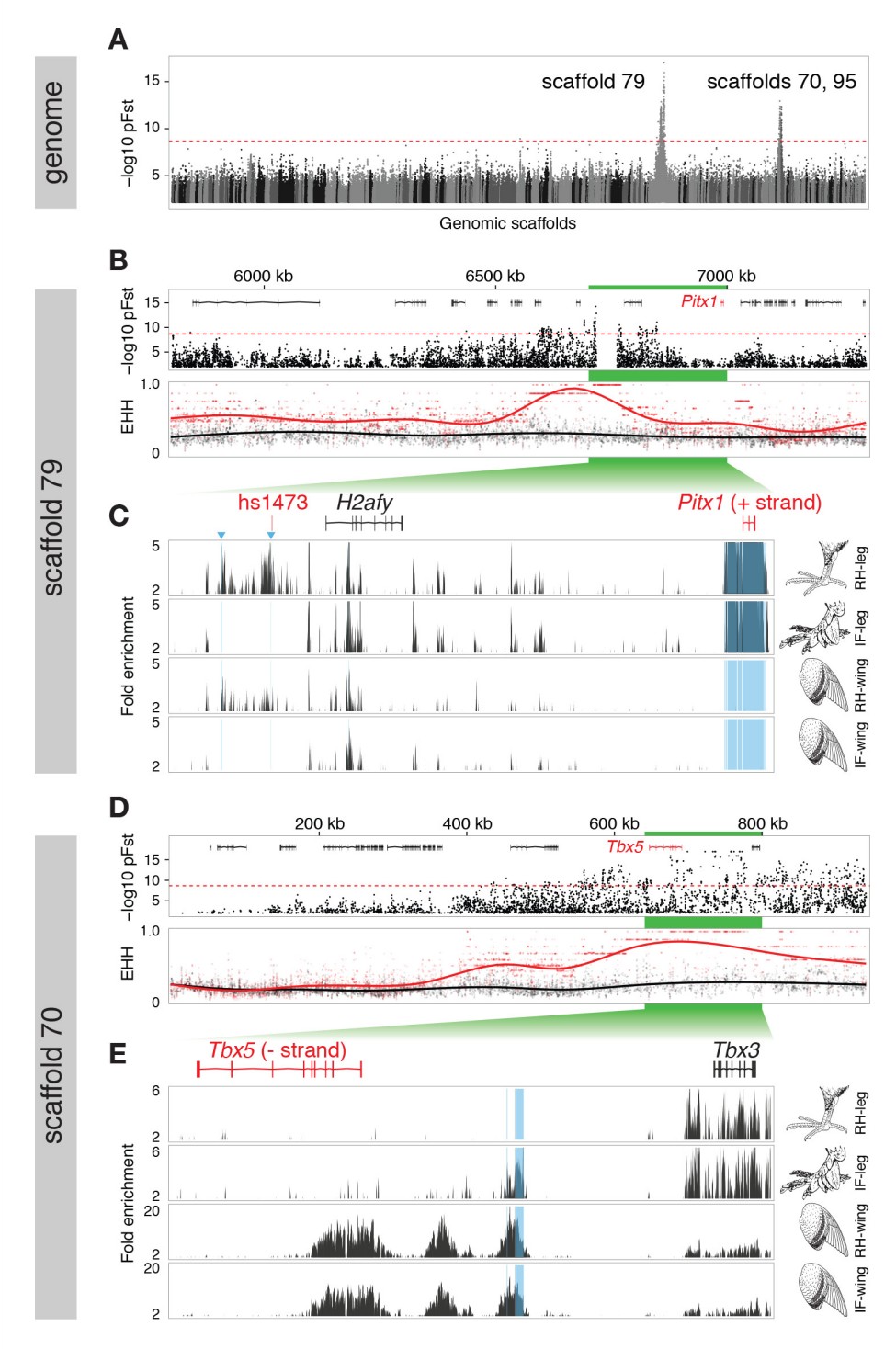

**Figure 2.** Two regions of genomic differentiation and H3K27ac enrichment distinguish scale- and feather-footed pigeons. (**A**) Whole-genome pFst comparisons between genomes of feather-footed and scale-footed pigeons. Scaffolds are ordered by genetic position in a linkage map from an $F_2$ cross (see **Figure 1**). Dashed line, genome-wide significance threshold. (**B**) pFst and extended haplotype homozygosity (EHH) plots for region of high differentiation on scaffold 79. Feather-footed birds (n=10, red in EHH plot) homozygous for a 44-kb deletion are differentiated from scale-footed birds (n=28, black) and show extended haplotype homozygosity in this region. Smoothed lines follow a generalized additive model (**Wickham, 2009**). (**C**) H3K27ac ChIP-seq enrichment differed significantly between embryonic wing and leg buds of the scale-footed racing homer (RH) in several regions (blue shading), including within a 44-kb interval that is deleted in the muffed Indian fantail (IF; blue arrowheads). This

*Figure 2 continued on next page*

*Figure 2 continued*
deleted region is orthologous to a known human limb enhancer (hs1473). (D) Selection scans show similar patterns of differentiation on scaffold 70 between muffed (n=11, red in EHH plot) and scale-footed birds (n=28, black). (E) H3K27ac ChIP-seq enrichment differed significantly between leg buds of racing homer and Indian fantail embryos in regions immediately 5' of *Tbx5* (blue shading). Foot and wing drawings modified after *Levi (1986)*.
The following figure supplements are available for figure 2:

**Figure supplement 1.** Synteny and genomic differentiation of pigeon LG20.
**Figure supplement 2.** Genomic association scans.
**Figure supplement 3.** Haplotype diagram of scaffold 70 candidate interval.

over this interval, suggesting that an additional feathered-foot allele may also exist at this locus (*Figure 2—figure supplement 2A*).

In contrast to the differentiation signal we observed between scale-footed and all feather-footed birds on scaffold 79, only the muffed birds (more heavily feathered) showed signatures of selection and shared similar haplotypes on scaffold 70 (higher pFst signal than the adjacent scaffold 95) (*Figure 2D*, *Figure 2—figure supplements 2C*, *3*). Thus, both QTL analyses and whole-genome scans show that mutation on scaffold 79 alone is sufficient for the grouse phenotype (*gr* locus), and point to scaffold 70 as the major-effect locus for longer toe feathers in birds with muffs (*Sl* locus) (*Figure 1F,G*).

## Expression of limb outgrowth and identity genes differs between scale-footed and feather-footed breeds

Next, we examined scaffolds 79 and 70 for candidate genes that might control the scale-to-feather transition. The highest pFst peak on scaffold 79 – corresponding to the major-effect QTL on LG 11 for the proportion of tarsometatarsal feathering – was approximately 200 kb upstream of *Pitx1*, a gene that encodes a homeobox-containing transcription factor that is normally expressed in the vertebrate hindlimb but not the forelimb (*Figure 2B*). The highest pFst peak on scaffold 70 – corresponding to the major-effect QTL on LG 20 for toe feather length – was 40 kb upstream of *Tbx5*, a gene that encodes a T-box transcription factor that is normally expressed in the vertebrate forelimb but not the hindlimb (*Figure 2D*). These regions were especially intriguing because these two genes encode key transcriptional regulators of forelimb (*Tbx5*) and hindlimb (*Pitx1*) identity and development (*Logan et al., 1998*; *Logan and Tabin, 1999*; *Rodriguez-Esteban et al., 1999*; *Szeto et al., 1999*; *Takeuchi et al., 1999*). For example, misexpression of *Pitx1* in the embryonic chick forelimb blocks feather development (*Logan and Tabin, 1999*), while misexpression of *Tbx5* in the early hindlimb field of embryonic chickens is sufficient to induce a partial wing-like transformation, including feather formation on the feet (*Takeuchi et al., 1999*). In mouse, *Pitx1* (but not *Tbx5*) plays a role in determining limb-type identity (*Szeto et al., 1999*; *Minguillon et al., 2005*; *DeLaurier et al., 2006*), suggesting that the roles of *Tbx5* in limb outgrowth and identity have diversified during amniote evolution (*Horton et al., 2008*).

We did not identify any fixed non-synonymous coding changes in *Pitx1* or *Tbx5* between scale-footed and feather-footed breeds of pigeon. However, we found striking differences in embryonic hindlimb expression of these genes among three different breeds – racing homer (scale-footed), Indian fantail (small-muffed), and English trumpeter (large-muffed) – at Hamburger-Hamilton stage 25 (HH25; *Hamburger and Hamilton, 1951*). *Pitx1* expression was reduced in both muffed breeds (expression relative to racing homer: Indian fantail 0.75 ± 0.06, p=0.0007; English trumpeter 0.40 ± 0.05, p=0.0007; n = 6 each) and was more severely reduced in the large-muffed English trumpeter (p=0.002) (*Figure 3A*). Conversely, *Tbx5*, the forelimb-specific transcription factor, was ectopically expressed in the hindlimbs of both muffed breeds (hindlimb expression relative to racing homer forelimb: racing homer 0.001 ± 0.0004; Indian fantail 0.01 ± 0.008, p=0.0007; English trumpeter 0.14 ± 0.05, p=0.0007; n = 6 each), and was higher in the large-muffed English trumpeter (p=0.002) (*Figure 3B*). Forelimb expression of *Tbx5* was indistinguishable among the three breeds, indicating

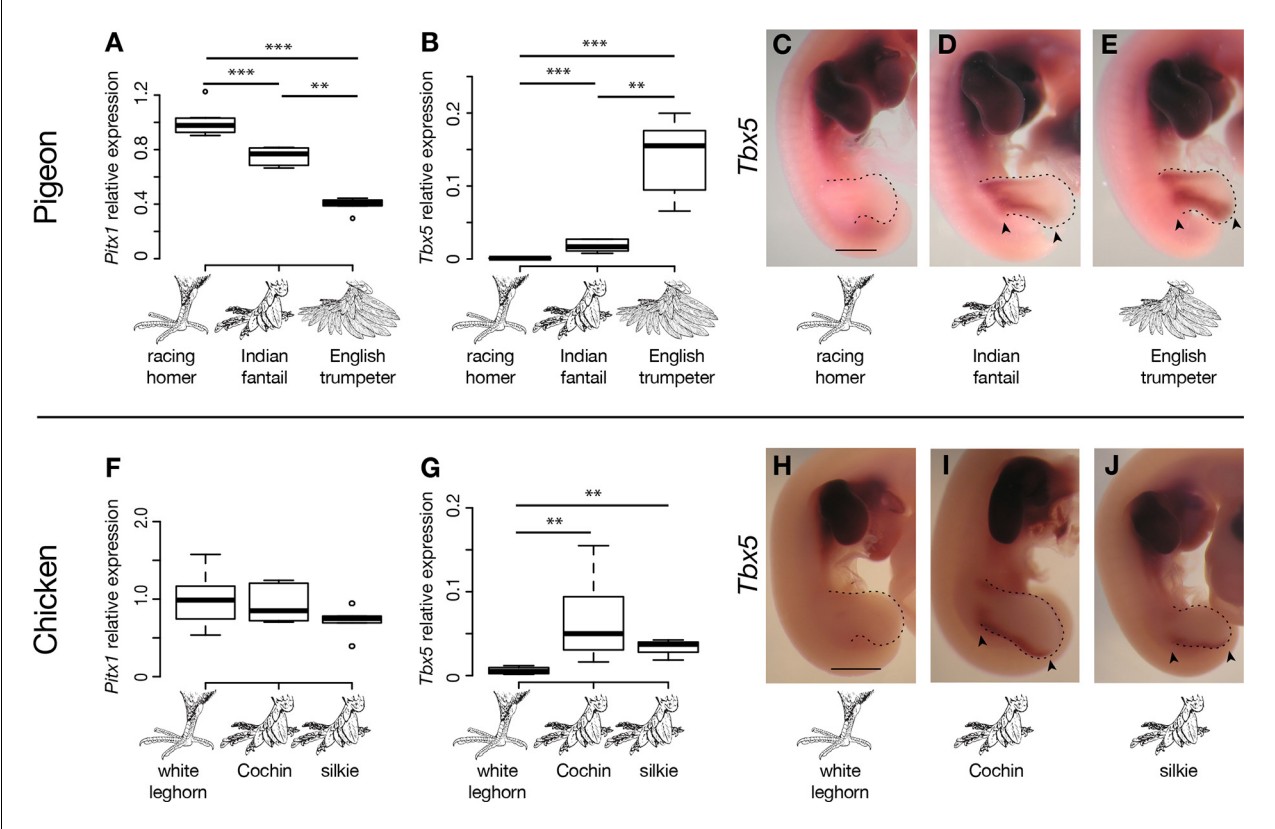

**Figure 3.** Limb-type gene expression varies among feathered and scaled pigeons and chickens. (A,B,F,G) qRT-PCR analyses of *Pitx1* and *Tbx5* expression in HH25 hindlimbs of pigeon (A,B) and chicken (F,G). Boxes span 1st to 3rd quartiles, bars extend to minimum and maximum observed values if within 1.5 times the interquartile range of the box, circles indicate values outside of this range, black line indicates median. **=p<0.01, ***=p<0.001. (C-E, H-J) RNA in situ hybridization for *Tbx5* expression in HH25 embryos of racing homer (C), Indian fantail (D), and English trumpeter (E) pigeons; and white leghorn (H), Cochin (I), and silkie (J) chickens. Arrowheads indicate sites of ectopic *Tbx5* expression. Scale bar = 1 mm.

The following source data and figure supplements are available for figure 3:

**Source data 1.** Source data from quantitative RT-PCR experiments.

**Figure supplement 1.** Quantitative RT-PCR expression analyses.

**Figure supplement 2.** Spatial expression pattern of *Pitx1* is similar in hindlimb buds of scaled-foot and feathered-foot embryos.

**Figure supplement 3.** Ectopic hindlimb expression of *Tbx5* and epidermal transformations in embryos and adults.

that upregulation of *Tbx5* in feather-footed breeds is restricted to the hindlimb (*Figure 3—figure supplement 1A*).

We examined expression of additional genes within the two candidate regions at HH25, and found that the *Tbx5* paralog *Tbx3* was also differentially expressed in both feather-footed pigeon breeds relative to racing homer (*Figure 3—figure supplement 1B*). This could be due to the fact that *Tbx3* is a target of *Tbx5* (*Mori et al., 2006*; *Postma et al., 2008*), and additional experiments confirm that *cis*-regulatory changes do not drive this expression difference (see below). The hindlimb-specific transcription factor *Tbx4* is not contained in the candidate regions defined by our QTL mapping and genome-wide association studies, but this gene is a downstream transcriptional target of Pitx1 (*Logan and Tabin, 1999*; *Takeuchi et al., 1999*; *Duboc and Logan, 2011*). We therefore compared expression levels of *Tbx4* among scaled and feathered breeds at HH25, but found no significant differences at this stage (racing homer 1.00 ± 0.21; Indian fantail 1.00 ± 0.17, p=0.85; English trumpeter 1.05 ± 0.19, p=0.66; n = 6 each) (*Figure 3—figure supplement 1C*). Thus, the

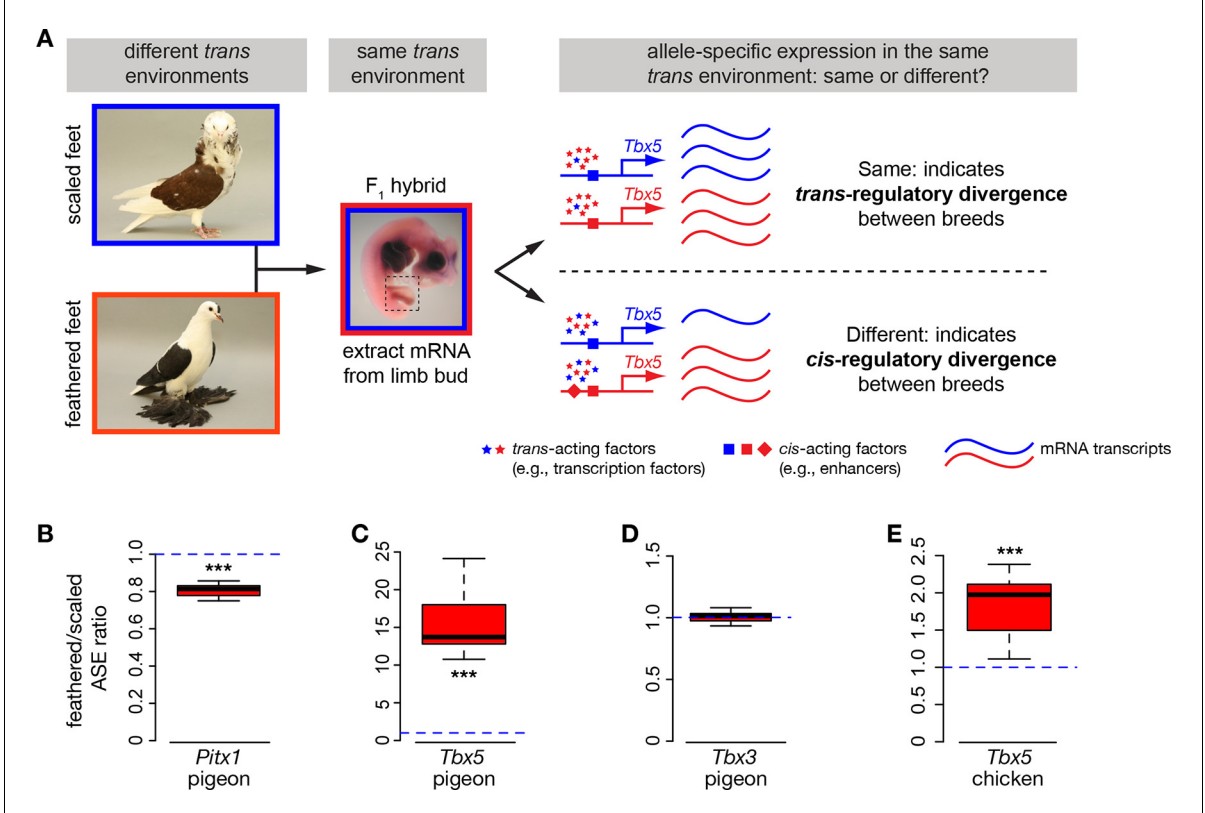

**Figure 4.** Allele-specific expression (ASE) assays demonstrate cis-regulatory changes in *Pitx1* and *Tbx5*. (**A**) Schematic of ASE assay using *Tbx5* expression as an example. Differences in *Tbx5* expression between scale-footed and feather-footed breeds could be due to *trans*- and/or *cis*-acting mutations. If expression differences between parental breeds are due to *trans* changes only (stars), then expression of the two *Tbx5* alleles in hybrid embryos will be the same (top right). In contrast, if *cis*-regulatory changes underlie differences in *Tbx5* expression between the parental breeds, then expression of the two *Tbx5* alleles in hybrid embryos will be different (bottom right). (**B-D**) ASE assay in hybrid hindlimb buds indicate *cis*-regulatory divergence between scale-footed (Old Dutch Capuchine) and muffed (fairy swallow) pigeon breeds in *Pitx1* (**B**) and *Tbx5* (**C**), but not in *Tbx3* (**D**). Dashed blue line indicates null hypothesis of equal expression of alleles. (**E**) ASE assay in hybrid hindlimb buds indicate cis-regulatory divergence in *Tbx5* between feather-footed (silkie) and scale-footed (white leghorn) chicken breeds. Boxes in (**B-E**) span 1st to 3rd quartiles, bars extend to minimum and maximum observed values if within 1.5 times the interquartile range of the box, circles indicate values outside of this range, black line indicates median. **$p \leq 0.01$ ***$p \leq 0.005$.

The following source data is available for figure 4:

**Source data 1.** Source data from pyrosequencing ASE experiments.

embryonic hindlimbs of muffed pigeons show quantitative expression changes in transcription factors with reciprocal limb expression domains, including the striking downregulation of a key hindlimb identity gene (*Pitx1*), and the novel expression of a key forelimb-specific gene (*Tbx5*).

We next analyzed the patterns of *Tbx5* and *Pitx1* expression in scaled and muffed pigeon embryos at HH25. Interestingly, ectopic hindlimb expression of *Tbx5* in muffed embryos was markedly different than its normal forelimb pattern in wild-type pigeons and other vertebrates (*Gibson-Brown et al., 1998a*; *1998b*; *Logan et al., 1998*; *Tamura et al., 1999*; *Ruvinsky et al., 2000*). *Tbx5* is typically expressed throughout the mesoderm of the forelimb, but ectopic *Tbx5* expression was largely localized to the mesoderm of the proximal and posterior-dorsal hindlimb of the small-muffed Indian fantail (*Figure 3D*, *Figure 3—figure supplement 3B*). This domain was further expanded in the large-muffed English trumpeter (*Figure 3E*, *Figure 3—figure supplement 3C*), consistent with the quantitative differences in expression between the two breeds (*Figure 3B*). This domain shows a striking correlation with regions of epidermal transformation, as foot feathers are usually longest and most numerous on the posterior digits (*Darwin, 1868*; *Levi, 1986*) (*Figure 3—*

figure supplement 3D–N). In contrast, and despite quantitative expression differences among breeds, *Pitx1* had a qualitatively similar expression domain in embryos of scale-footed and feather-footed breeds at this stage (*Figure 3—figure supplement 2*). Therefore, consistent with the critical role of mesoderm in determining ectodermal fate (*Hughes et al., 2011*), regionalized ectopic expression of *Tbx5* is correlated with enhanced local transformation of epidermal appendages.

## *Cis*-regulatory changes contribute to expression differences in *Pitx1* and *Tbx5* in muffed pigeons

If *cis*-acting regulatory mutations are responsible for the differences in *Pitx1*, *Tbx5*, and/or *Tbx3* expression between embryos of scale-footed and feather-footed pigeons, then differential expression of scaled-foot and feathered-foot alleles should persist in a common *trans*-acting cellular environment. To test this prediction, we generated $F_1$ hybrid pigeon embryos by crossing an Old Dutch Capuchine (scale-footed) to a fairy swallow (muffed), and measured parent-of-origin allele expression in the hybrid embryonic hindlimb at HH25 (*Figure 4A*) (*Domyan et al., 2014*). Consistent with expression differences we observed *among* breeds (*Figure 3A,B*), expression of the feathered-foot allele of *Pitx1* was approximately 20% lower than the scaled-foot allele (expression of feathered-foot relative to scaled-foot allele: $0.807 \pm 0.039$, p=0.003, n = 6 embryos), and expression of the feathered-foot allele of *Tbx5* was nearly 1600% higher than the scaled-foot allele (relative expression of feathered-foot allele: $15.75 \pm 4.69$, p=0.002, n = 7 embryos) (*Figure 4B,C*). In contrast, expression levels of feathered-foot and scaled-foot alleles of *Tbx3* were indistinguishable (relative expression of feathered-foot allele: $0.99 \pm 0.05$, p=0.68, n = 7 embryos) (*Figure 4D*). These results directly show that *cis*-acting genetic changes alter expression of feathered-foot alleles of *Pitx1* and *Tbx5*, but not *Tbx3*, in the embryonic pigeon hindlimb.

## Patterns of open chromatin differ between scale-footed and feather-footed pigeon breeds

We next performed genome-wide comparisons of differential enhancer activity in embryonic limbs of racing homers and Indian fantails, using H3K27ac as a marker for open chromatin (*Creyghton et al., 2010*). Strikingly, two regions of significantly different enrichment in racing homer hindlimbs relative to forelimbs were within the 44-kb genomic region that is deleted in the Indian fantail, and one of these regions (log10 likelihood ratio = 4.93) is adjacent to the hs1473 limb enhancer (*Figure 2C*). Furthermore, the most significant differentially enriched region in Indian fantail hindlimbs relative to racing homer hindlimbs was directly upstream of *Tbx5* (log10 likelihood ratio = 10.3) (*Figure 2E*). The overlapping patterns of enrichment in Indian fantail hindlimbs and wild-type forelimbs suggest that ectopic hindlimb expression of *Tbx5* could be due to de-repression of forelimb-specific enhancers. In summary, the differential expression of *Pitx1* and *Tbx5* among pigeon breeds (*Figure 3*) and between alleles (*Figure 4*) is also reflected by differential chromatin activation at these genes.

## Muffed pigeon breeds incur musculoskeletal patterning changes

In mouse and chicken embryos, experimental manipulation of *Pitx1* and *Tbx5* expression can result in muscular and skeletal abnormalities. Experiments in both chick and mouse consistently demonstrate that *Pitx1* plays a necessary (but not sufficient) role in determining hindlimb-type morphology of the skeleton, muscles, and tendons (*Logan and Tabin, 1999*; *Takeuchi et al., 1999*; *Minguillon et al., 2005*; *DeLaurier et al., 2006*; *Duboc and Logan, 2011*). Experimentally induced ectopic expression of *Tbx5* in the hindlimbs of chick embryos can also induce muscular and skeletal anomalies, although *Tbx5* does not directly control limb skeletal patterning or determine forelimb-type morphology in mice (*Rodriguez-Esteban et al., 1999*; *Takeuchi et al., 1999*; *Minguillon et al., 2005*; *Hasson et al., 2007*) However, normal patterning of limb muscles and tendons is dependent on *Tbx5* and *Tbx4* in mice (*Hasson et al., 2010*). These apparent discrepancies between mammalian and avian systems point to subtle differences in limb development in different lineages (*Horton et al., 2008*).

Given the dramatic musculoskeletal defects observed in other organisms with experimentally altered *Pitx1* and *Tbx5* expression, we compared the hindlimb morphology of adult feral pigeons (scale-footed, n=2) to that of the English trumpeter (muffed, n=2) and the Pomeranian pouter

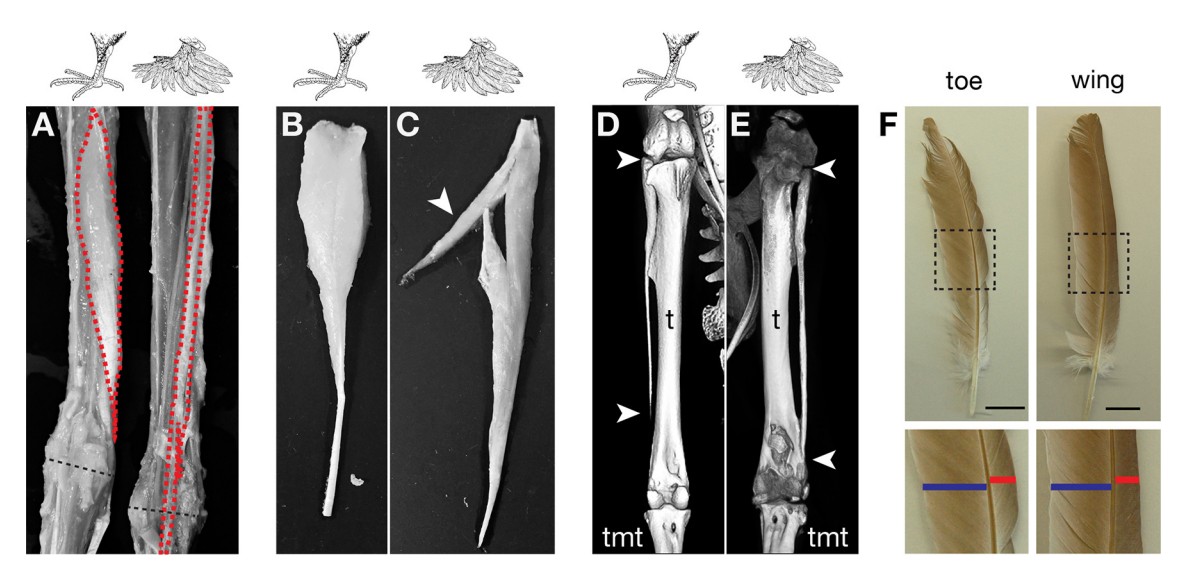

**Figure 5.** Muffed pigeons have re-patterned hindlimb musculoskeletal system and wing-like foot feathers. (**A,B,C**) Gross muscle morphology of scale-footed (feral) and muffed (Pomeranian pouter) left hindlimbs, dorsal view. (**A**) Skin and superficial muscles have been removed to reveal re-patterning of the fibularis longus (FL, red). Dashed black line, approximate position of ankle joint axis. (**B,C**) The FPP3 is a pinnate muscle in scale-footed pigeons (**B**), but a slip of fibers fuses with the adjacent FL in muffed pigeons (arrowhead in **C**). (**D,E**) X-ray computed tomography images of scale-footed (feral, right leg) and muffed (English trumpeter, left leg) hindlimbs. Arrowheads mark the proximal and distal ends of the fibula. The wild-type pigeon fibula (**D**) is short and splint-like. In the muffed bird (**E**), the fibula extends from the knee to the ankle. We observed distal elongation of the fibula in another muffed breed (fairy swallow) but the fibula did not completely extend to the ankle (not shown). t, tibia; tmt, tarsometatarsus. (**F**) Toe and wing (flight) feathers of a muffed pigeon (English trumpeter), highlighting vane width asymmetries. Blue bar, inner vane; red bar, outer vane. Scale bar = 2 cm.

(muffed, n=1). We found consistent soft-tissue patterning defects in both feather-footed pigeon breeds: the fibularis longus (FL) tendon inserts on the dorsal rather than ventral surface of the proximal tarsometatarsus, the flexor perforans et perforatus (FPP3) muscle adopts a longitudinal rather than pennate fiber orientation, and a slip of the FPP3 fuses with the FL tendon (*Figure 5A*). These changes are aberrations of normal patterning, although they are not necessarily clear transformations to a more forelimb-like configuration. We also found that the fibula, which is normally splint-like and shorter than the tibia in pigeons, was enlarged (*Figure 5D,E*) and two phalanges of digit 4 were fused in feather-footed breeds (not shown). These are not necessarily limb-type transformations, either. However, experimental ectopic expression of *Tbx5* in the hindlimbs of chick embryos produces an enlargement of the fibula reminiscent of extreme pigeon phenotypes, and *Takeuchi et al., 1999* compared this morphology to a forelimb-like condition (the fibula "makes a joint at its distal end like a normal ulna [the corresponding postaxial zeugopod bone of the forelimb]," p. 810). Notably, all of the modified structures of ptilopodous pigeons develop in the posterior (lateral in the adult) and dorsal hindlimb, which are the primary sites of ectopic *Tbx5* expression. Thus, the morphological changes to the hindlimbs of feather-footed pigeon breeds are considerably more than skin deep.

### *Tbx5* is ectopically expressed in the hindlimb buds of feather-footed chickens

Other bird species, including domestic chickens, independently evolved foot feathers. Similar to pigeons, *Tbx5* was ectopically expressed at HH25 in hindlimb buds of two feather-footed chicken breeds, the Cochin and the silkie (hindlimb expression relative to white leghorn forelimb: white leghorn $0.005 \pm 0.004$; Cochin $0.066 \pm 0.050$, p=0.002; silkie $0.034 \pm 0.014$, p=0.009; n = 6 white leghorns, 6 Cochins, 4 silkies) (*Figure 3G*). Ectopic *Tbx5* expression in feathered-foot chicken embryos had a similar domain to that of feathered-foot pigeon embryos at HH25 (*Figure 3I,J*) and, as in pigeons, *cis*-acting changes contributed to this expression (expression of feathered-foot allele relative to scaled-foot allele in HH25 silkie x white leghorn $F_1$ hybrid hindlimbs: $1.80 \pm 0.41$, p=2.55 x

$10^{-5}$, n = 11 hybrid embryos) (*Figure 4E*). Hence, *Tbx5*-related developmental mechanisms may, in part, underlie the evolution of foot feathering in two species that last shared a common ancestor over 80 million years ago (*Claramunt and Cracraft, 2015*).

Classical genetic studies implicate at least two loci in heavy foot feathering in chickens (*Punnett and Bailey, 1918*; *Lambert and Knox, 1929*; *Warren, 1948*; *Somes, 1992*), although the molecular genetic origins of the trait remain unknown. Previously, a chromosome region containing *Pitx1* was implicated in foot feathering in silkie chickens (*Dorshorst et al., 2010*). However, we did not detect statistically significant changes in *Pitx1* expression between scaled-foot (white leghorn) and feathered-foot (silkie and Cochin) chicken embryos at HH25 (expression relative to white leghorn: Cochin 0.92 ± 0.24, p=0.93; silkie 0.71 ± 0.18, p=0.18; n = 6 each) (*Figure 3F*). This apparent conflict could be because the causative gene in silkies is not actually *Pitx1* but rather a gene closely linked to it, or because *Pitx1* expression differences are more pronounced and consistent at developmental stages that we did not assay. Furthermore, different populations of breeds such as silkies appear to have different constellations of ptilopody loci and alleles, and it is possible that we used strains that do not have *Pitx1* mutations (*Wexelsen, 1934*; *Somes, 1992*). Also in contrast to our results in feather-footed pigeons, *Tbx3* was not upregulated in ptilopodous chicken breeds (white leghorn 1 ± 0.17, silkie 0.48 ± 0.18, p=0.004; Cochin 0.86 ± 0.44, p=0.40; silkie vs. Cochin p=0.07; n = 6 samples each) (*Figure 3—figure supplement 1D*). In all, these results suggest that both shared and distinct mechanisms regulate foot feathering among avian species.

## Discussion

### Genetic architecture of ptilopody in pigeons

Extensive classical breeding experiments in pigeons demonstrate that complex derived traits can often be parsed into component parts (*Sell, 1994*, *2012*). Thus, while traits are not always simple, they are often genetically tractable when using an informed breeding strategy (*Domyan et al., 2014*). Equivalent insights about the genetic architecture of phenotypic divergence between wild vertebrate species are often considerably more difficult to acquire. With pigeons, however, we have documentation for specific breed selection criteria and direct evidence for the resulting genetic architecture of derived traits (*Levi, 1965*; *1986*; *Sell, 1994*; *National Pigeon Association, 2010*; *Sell, 2012*). This information offers a crucial advantage because it informs how we design genetic crosses and choose breeds for whole-genome resequencing to identify causal genes and mutations. Thus, we can combine classical breeding strategies and genomics to identify the molecular basis of both simple and oligogenic traits, as well as dissect different components of a complex phenotype, and define functional interactions among genes (*Shapiro and Domyan, 2013*; *Domyan et al., 2014*; *Vickrey et al., 2015*).

Classical studies in pigeon suggest two major-effect loci – *grouse* (*gr*) and *Slipper* (*Sl*) – are responsible for most of the variation in foot feathering (*Doncaster, 1912*; *Wexelsen, 1934*; *Hollander, 1937*; *Levi, 1986*). Through a combination of genetic, genomic, and developmental approaches, our data implicate regulatory mutations in the limb outgrowth and identity genes *Pitx1* and *Tbx5* as the molecular identities of the *gr* and *Sl* locus, respectively (*Figure 6*). Each locus has significant and separable effects on qualitative and quantitative variation in epidermal appendages: derived alleles of *Pitx1* increase the extent of foot feathering, while a derived allele of *Tbx5* is associated with the more elaborate muffed phenotype (*Figure 1*). Notably, these feathers are most numerous on the central forward-facing toe (digit 3; *Table 2*; *Figure 3—figure supplement 3*), just as forelimb feathers are most numerous on the central forelimb digit in birds and their dinosaurian relatives (*Yalden et al., 1985*; *Gishlick et al., 2001*; *Hieronymus, 2015*). Further, we also find that muscular and skeletal morphology are altered in muffed pigeons.

Collectively, these findings point to a partial alteration of the identity of the developing hindlimb, rather than localized changes to individual epidermal placodes. These alterations do not represent a complete transformation of limb type, as the hindlimbs of feather-footed pigeons are still readily recognizable as legs. This suggests that limb-type identity is not a simple binary choice between two global fates. For example, feather-footed pigeons have a radical transformation of the distal hindlimb dermis, yet changes to other hindlimb mesoderm derivatives (muscle, skeleton) are subtler and largely restricted to lateral structures in the adult. Therefore, we propose that different aspects

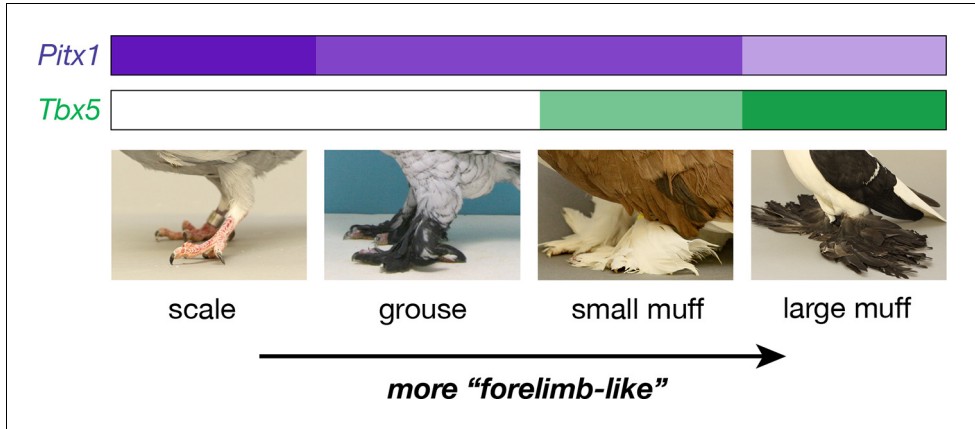

**Figure 6.** Model describing link between *Pitx1* and *Tbx5* expression levels and foot epidermal appendage morphology. Darker colors indicate higher expression levels. Decreased expression of *Pitx1* and ectopic expression of *Tbx5* are associated with foot feathering (and other morphological transformations) in domestic pigeons.

of fore- and hindlimb morphology could have different dosage- and/or stage-dependent requirements for exposure to identity cues. Our ongoing analyses of musculoskeletal phenotypes in our F$_2$ cross, which includes individuals with different combinations of feathered-foot alleles of *Pitx1* and *Tbx5*, will allow us to understand the separate and epistatic effects of these loci on musculoskeletal anatomy. We note that, although genetic manipulations indicate that *Tbx5* does not specify forelimb identity in mice, the divergence time between mammals and birds is deep (>300 million years) and subtly different roles for this transcription factor in limb outgrowth and identity might have evolved in these lineages (*Minguillon et al., 2005*; *Horton et al., 2008*).

### Roles of *Pitx1* and *Tbx5* in diversity and disease

Another important avenue of future research will be to determine the downstream molecular consequences of *Pitx1* and *Tbx5* misregulation, and how this ultimately results in the transformation of scaled into feathered epidermis. Mutations at these genes can cause congenital limb deformities in humans, including clubfoot and Liebenberg syndrome (*Pitx1*; *Gurnett et al., 2008*; *Spielmann et al., 2012*), and Holt-Oram syndrome (*Tbx5*; *Basson et al., 1997*). Notably, haploinsufficiency causes these human syndromes, and clubfoot is partially penetrant in *Pitx1*$^{+/-}$ mice (*Alvarado et al., 2011*), collectively pointing to an exquisite sensitivity of limb morphology to levels of *Pitx1* and *Tbx5* gene products. *Pitx1* is also involved repeatedly in the evolution of adaptive pelvic fin loss in stickleback fish (*Cresko et al., 2004*; *Shapiro et al., 2004*; *2006*; *Coyle et al., 2007*; *Chan et al., 2010*; *Shikano et al., 2013*). Threespine sticklebacks that are homozygous for a *Pitx1* pelvic enhancer deletion have severely reduced or absent pelvises, but heterozygotes also have smaller pelvises, thereby allowing natural selection to act on fish carrying one mutant allele (*Shapiro et al., 2004*; *Chan et al., 2010*). Similarly, we observed increased foot feathering in pigeons with just one derived allele of *Pitx1* or *Tbx5* (*Figure 1*). Ancient pigeon breeders could have rapidly selected for ptilopodous pigeon lines starting with birds that were heterozygous for mutations at either locus, and later generated the extreme muffed phenotype by hybridization. Together, studies of diversity and disease indicate that modest changes to the amount and location of *Pitx1* and *Tbx5* gene expression can cause dramatic alterations to limb development and morphology.

In addition to implicating *cis-* acting mutations in *Pitx1* and *Tbx5* driving transformation of scales into feathers in domestic pigeon, our results suggest that additional, as yet unidentified, mutations contribute to the muff phenotype. Although all feathered-foot embryos examined in our gene expression experiments contained derived *Pitx1* and *Tbx5* haplotypes, misregulation of each gene was more severe in the large-muffed English trumpeter than in the small-muffed Indian fantail. The English trumpeter may therefore contain additional *cis*-acting mutations at one or both loci, and/or mutations in upstream regulators of *Pitx1* and *Tbx5*. Additional studies will be required to

discriminate between these possibilities. Our findings suggest a quantitative link between transcription factor abundance and skin appendage fate and morphology, thereby highlighting foot-feathering in pigeons as a model for studying the regulatory interactions that govern expression of these two important determinants of limb morphology.

## Evolution of epidermal appendage distribution

How might pigeons help us understand the evolution of epidermal appendage distribution and limb morphology in other species? Our findings suggest the mechanistic basis for the development of feathered feet in two distantly related domestic bird species is due to a partial transformation of limb identity, through *cis*-acting regulatory mutations in limb-type specific transcription factors. Most modern wild birds have a scaled metatarsus and toes, although some species (e.g., ptarmigan, snowy owl, and golden eagle) have extensive foot feathering. However, recent paleontological evidence suggests that feathers – not scales – might be the ancestral hindlimb skin appendages in birds and some of their close non-avian dinosaur relatives (*Hu et al., 2009*; *Zheng et al., 2013*). Thus, the epidermis of feather-footed modern birds might actually represent a reversion to the ancestral avian skin condition. In some cases, the large, asymmetric-vaned, pennaceous metatarsal feathers of basal birds and their non-avian dinosaur relatives are so extensive that their hindlimbs have been interpreted as 'hind wings', although they clearly retain hindlimb skeletal identity (*Zheng et al., 2013*). Furthermore, these long foot feathers are directed laterally from the foot, and they display vane width asymmetries reminiscent of flight feathers; we find a similar morphology in muffed pigeons (*Figure 5F*, *Figure 3—figure supplement 3*). Perhaps not coincidentally, *Darwin, 1868* noted of the muffed English trumpeter pigeon, "Their feet are so heavily feathered, that they almost appear like little wings" (p. 155).

Building on classical breeding experiments in both pigeons and chickens, we find that a relatively small number of genetic changes account for a large proportion of the variation in epidermal appendage morphology and distribution. Thus, major determinants of dramatic phenotypic variation can be mechanistically simple and therefore potentially evolve rapidly. In pigeons, these mechanisms can generate wing-like feathers on a hindlimb that is not used for powered flight or gliding. This, in turn, suggests that wing-like foot and leg feathers in other species, such as non-avian dinosaurs, might result from developmental constraints on the morphology of large limb feathers, rather than from functional adaptations for flight (*Gould and Lewontin, 1979*; *Foth et al., 2014*).

## Materials and methods

### Animal husbandry and phenotyping of F$_2$ offspring

Animals were housed in accordance with the University of Utah Institutional Animal Care and Use Committees of University of Utah (protocols 10–05007 and 13–04012). 130 F$_2$ offspring were generated by mating a male Pomeranian pouter to two female Scandaroons, and DNA samples extracted (DNeasy Blood and Tissue Kit, Qiagen, Valencia, CA). 114 F$_2$ offspring survived to 6 months of age, at which time they were euthanized and phenotypic measurements taken. Proportion of the tarsus was measured by dividing the length of the dorsal tarsus that was feathered by the total length of the tarsus (measured from the tibia-tarsometatarsus joint to the distal aspect of tarsometatarsal-phalangeal joint of digit 3), and averaged between the two tarsi. Toe feathers were counted on each toe, and summed across all 8 toes. The length of each of the longest three toe feathers on digit 3 (the central forward-directed toe), which bore the longest toe feathers on each foot, was measured to the nearest 1 mm and averaged for each bird.

### Whole-genome genotyping by sequencing (GBS)

For genotyping, we used a previously published approach (*Elshire et al., 2011*) with minor modifications. Briefly, for each founder parent and 130 F$_2$ offspring, 50 ng of DNA was digested with ApeKI, ligated to barcoded adapters, and then 10 ng of each barcoded sample was pooled in batches of 26 individuals and purified (Qiagen PCR Purification Kit). DNA fragments 550–650 bp in size were selected using Pippin Prep (Sage Science, Beverly, MA), and amplified by 10–12 cycles of PCR using custom indexed primers. Libraries were purified with Ampure beads (Sigma-Aldrich, St. Louis, MO) and sequenced using 100- or 125-bp, paired-end sequencing on the Illumina HiSeq2000 platform at

the University of Utah Genomics Core Facility. Reads were trimmed to 90 bp, filtered for quality, and de-multiplexed using Stacks (*Catchen et al., 2011*). Reads were mapped to the pigeon reference genome (*Shapiro and Domyan, 2013*) using Bowtie2 (*Langmead and Salzberg, 2012*), filtering for MAPQ < 20. The average number of mapped reads among $F_2$ individuals was 3,397,598, with a mean depth of 6.3x. Genotypes were called using Stacks (*Catchen et al., 2011*), with a minimum read-depth cutoff of 5. Markers that were genotyped in $\geq$ 70 of the 130 $F_2$ individuals were retained.

## Genetic map construction and QTL mapping

Genetic map construction and QTL mapping was performed using R/qtl (www.rqtl.org) (*Broman et al., 2003*). Markers showing segregation distortion (Chi-square, p<0.05) were removed. 3803 markers were assembled into linkage groups using the parameters (max.rf = 0.15, min.lod = 6). Linkage groups were numbered in descending order, based on the number of markers. Linkage group 11 and 20 QTL were initially mapped using the *scanone* function using Haley-Knott regression. Probable false-homozygote genotyping errors resulting from the low read-depth cutoff used (5x), identified as closely-spaced double-crossover events, were manually corrected on these linkage groups. Subsequently, the *stepwiseqtl* function was used to identify additional QTL, and the *fitqtl* function used to account for the effect of one linkage group while calculating the LOD scores and percent variance explained (PVE) of the other. Significance thresholds of $\alpha$ = 0.05 were calculated with 1000 permutations of each phenotype across all linkage groups. The peak markers for each phenotype were used to test for the effect of each QTL.

## Genomic analyses

BAM files generated previously for a whole-genome resequencing panel (*Shapiro and Domyan, 2013*) were combined with BAM files for two new Pomeranian pouter whole-genome sequences to call genomic variants (SNVs and small indels) using the Genome Analysis Toolkit (Unified Genotyper and LeftAlignAndTrimVariants functions, default settings) (*McKenna et al., 2010*). We removed variant sites that were called in two or fewer genomes (i.e., all other genomes were no-calls) or that had variant alleles on only two or fewer chromosomes, as these categories of low-frequency variants would be uninformative to our analyses. The resulting variant call format (VCF) file was used for subsequent analyses.

Individual birds from different breeds were binned into the following phenotypic classifications:

*Groused*: Berlin long-faced tumbler, Lahore, Oriental frill, Shaksharli.

*Muffed:* English long-faced muffed tumbler, English pouter, English trumpeter, frillback, ice pigeon, Indian fantail (2 individuals), Pomeranian pouter (2 individuals), Saxon monk, Saxon pouter. The English pouter is an unusual breed that is sometimes classified by breeders as slipper only. Its foot feathering is far more extensive than groused breeds, which led us to include it in the muffed group for the purposes of the genomic analyses.

*Scale-footed:* African owl, archangel, Birmingham roller, carneau, Chinese owl, cumulet, Egyptian swift, English carrier, fantail, feral (2 individuals), Iranian tumbler, Jacobin, king, Lebanon, Marchenero pouter, mookee, Oriental roller, parlor roller, runt, Scandaroon, Spanish barb, starling, Syrian dewlap, Thai laugher.

pFst, a modified likelihood ratio test that accounts for genotype uncertainty, extended haplotype homozygosity (EHH), and haplotype network analyses were implemented using the GPAT++ software library (*Kronenberg et al., 2014*; see https://github.com/vcflib/vcflib for software updates).

## Genotyping assays

Primers for genotyping the scaffold 79 deletion are listed in *Supplementary file 1*. Breeds used for association testing were as follows:

*Feather-footed* (21 breeds total): Berlin long-faced tumbler, Berlin short-faced tumbler, Bokhara trumpeter, classic Oriental frill, crested Saxon field color, English trumpeter, fairy swallow, frillback, German double-crested trumpeter, ice pigeon, Indian fantail, Lahore, Mindian fantail, Oriental frill, Persian roller, Pomeranian pouter, Russian tumbler, saint, Schmalkaldner moorhead, Uzbeck tumbler, West of England.

*Scale-footed* (56 breeds total): African owl, Altenburg trumpeter, American flying tumbler, American giant homer, American mini crest, American show racer, archangel, Bohemian pouter, Brunner cropper, Budapest tumbler, Cauchois, Chinese owl, cumulet, Danzig highflier, domestic show flight, dragoon, English baldhead long-faced clean-legged tumbler, English carrier, English magpie, English short-faced tumbler, exhibition homer, fantail, Franconian trumpeter, French mondaine, giant runt, Holle cropper, horseman pouter, Italian owl, Jacobin, Jiennesse pouter, king, Lebanon, Spanish little friar tumbler, Maltese, medium-faced crested helmet, Modena, mookee, Norwich cropper, nun, Old Dutch Capuchine, Old German owl, Oriental roller, parlor roller, Portuguese tumbler, Scandaroon, showtype racing homer, Spanish barb, starling, Syrian Baghdad, Texas pioneer, Thai laugher, Thuringer clean leg, Vienna medium-faced tumbler, Voorburg shield cropper, zitterhals.

## High-throughput chromatin immunoprecipitation and sequencing (ChIP-seq)

Forelimb and hindlimb buds from HH25 racing homer and Indian fantail embryos were collected and placed in 1% formaldehyde for 20 min at room temperature, then washed 3x in ice-cold PBS and stored at -80°C until chromatin extraction. ChIP was performed on 200 micrograms of chromatin isolated from embryonic pigeon limbs. Control libraries were prepared using 100 ng of input chromatin. A total of 16 libraries were created (8 ChIP and 8 input controls for each breed and limb combination). A validated monoclonal antibody against H3K27ac (Millipore #05–1334, Billerica, MA) was used to perform ChIP, and sequencing libraries were prepared using NEBNext Ultra DNA Library Prep Kit for Illumina with NEBNext Multiplex Oligos for Illumina (Index Primers Set 1; New England BioLabs, Ipswich, MA). All libraries were size selected using SPRI magnetic beads to eliminate adapter dimers. All 8 ChIP libraries showed enrichment for a positive control site relative to input libraries (tested by qPCR). Single-end, 50-bp read sequencing was performed on Illumina HiSeq2000 platform at the University of Utah Genomics Core Facility.

Fold-enrichment plots were generated using MACS (*Zhang et al., 2008*; *Feng et al., 2012*) and visualized in IGV (*Robinson et al., 2011*; *Thorvaldsdóttir et al., 2013*). Regions of differential enrichment between racing homer and Indian fantail hindlimbs were identified using function *bdgdiff* in MACS2 (https://pypi.python.org/pypi/MACS2/2.0.10.20130522). Regions with a log10 likelihood ratio $\geq$ 3 were considered to have differential enrichment between the two groups.

## RNA isolation and cDNA synthesis

To assay gene expression, limb buds from HH25 embryos were harvested and placed in RNAlater (Qiagen, Valencia, CA) at 4°C overnight. Total RNA was extracted, cleaned and DNase-treated (Qiagen RNeasy Kit). mRNA was reverse-transcribed to cDNA using oligo-dT and M-MLV RT (Invitrogen, Carlsbad, CA) according to the manufacturer's protocol.

## qRT-PCR analyses

cDNA was amplified using intron-spanning primers for each target using a CFX96 qPCR instrument and iTaq Universal Sybr Green Supermix (Bio-Rad, Hercules, CA). Results were compared by Mann-Whitney U test. Two technical replicates of each sample were performed, and the mean value determined. Differences were considered statistically significant if $p < (0.05 \ / \ \#$ genes assayed) to control for multiple-testing. Each experiment was performed three times, and the results presented are from one representative experiment. Primers used for each assay are listed in *Supplementary file 1*.

## Allele-specific expression assay

SNPs in *Pitx1* and *Tbx5* transcripts were identified by Sanger sequencing in the parents of a cross between an Old Dutch Capuchine (scale-footed) and a fairy swallow (muffed) that was homozygous for the 44-kb deletion upstream of *Pitx1*, and PyroMark Custom Assays (Qiagen) for each SNP were designed using the manufacturer's software. Pyrosequencing was performed on cDNA and gDNA derived from HH25 limb buds using a PyroMark Q24 instrument (Qiagen). The signal intensity ratio of feathered allele to scaled allele from cDNA samples was normalized to ratios obtained from gDNA samples from the same embryos to control for allele-specific amplification bias. Normalized ratios were analyzed by Mann-Whitney U test, and considered significant if $p < (0.05 \ / \ \#$ genes

assayed) to control for multiple-testing. Each experiment was performed twice, and the results presented are representative. Primers used for each assay are listed in *Supplementary file 1*.

### Whole-mount in situ hybridization

Linear templates for probe synthesis were amplified from cDNA by PCR using primers listed in *Supplementary file 1*. Binding sites for T3 and T7 polymerase were incorporated into the forward and reverse primers to facilitate subsequent transcription of sense and antisense probe, respectively.

Embryos used for RNA in situ hybridization were dissected from eggs, and fixed overnight in 4% paraformaldehyde at 4°C on a shaking table, then dehydrated into 100% MeOH and stored at -20°C. RNA in situ hybridization was performed as described (*Abler et al., 2011*). Hybridization with sense probe was performed as negative control.

## Acknowledgements

We thank Della Fixsen, Cassandra Garner, Kamala Ganesh, Patrick Miller, the University of Utah Genomics Core, and the University of Utah Small Animal Imaging Core for technical assistance, and members of the Utah Pigeon Club and National Pigeon Association for generously providing samples. We also thank Elena Boer for comments and discussion on the manuscript. Wild-type pigeon image in *Figure 5D* is courtesy of Dr. M Scott Echols, Grey Parrot Anatomy Project, University of Utah and the Medical Center for Birds (Oakley, California). This work was supported by the National Science Foundation (CAREER DEB1149160 to MDS, CAREER IOS1149453 to DBM, IOS0955517 EDEN internship to AIV), a Burroughs Wellcome Career Award in the Biomedical Sciences (MDS), and the National Institutes of Health (R01GM115996 to MDS, F32GM103077 and T32HD07491 fellowships to ETD, T32GM007464 fellowships to ZK and SAS, R01HD053728 to GK, R01GM104390 to MY). We acknowledge a computer time allocation from the Center for High Performance Computing at the University of Utah. Illumina shotgun reads for two Pomeranian pouter pigeons are deposited in NCBI BioProject ID PRJNA284526. H3K27ac ChIP-seq data are deposited in the Gene Expression Omnibus (GEO accession number GSE67875). Correspondence should be addressed to MDS (shapiro@biology.utah.edu).

## Additional information

### Funding

| Funder | Grant reference number | Author |
|---|---|---|
| National Institutes of Health | F32GM103077 Fellowship | Eric T Domyan |
| National Institutes of Health | T32HD07491 Fellowship | Eric T Domyan |
| National Institutes of Health | T32GM007464 Fellowships | Zev Kronenberg Sydney A Stringham |
| National Science Foundation | EDEN RCN Internship IOS0955517 | Anna I Vickrey |
| National Science Foundation | Graduate Research Fellowship | Rebecca Bruders |
| National Institutes of Health | R01HD053728 | Gabrielle Kardon |
| National Science Foundation | CAREER IOS1149453 | Douglas B Menke |
| National Institutes of Health | R01GM104390 | Mark Yandell |
| National Institutes of Health | R01GM115996 | Michael D Shapiro |
| Burroughs Wellcome Fund | Career Award in the Biomedical Sciences | Michael D Shapiro |
| National Science Foundation | CAREER DEB1149160 | Michael D Shapiro |

The funders had no role in study design, data collection and interpretation, or the decision to submit the work for publication.

## Author contributions

ETD, CRI, SP, MDS, Conception and design, Acquisition of data, Analysis and interpretation of data, Drafting or revising the article; ZK, Conception and design, Acquisition of data, Analysis and interpretation of data, Drafting or revising the article, Contributed unpublished essential data or reagents; AIV, SAS, RB, MWG, GK, Acquisition of data, Analysis and interpretation of data, Drafting or revising the article; JP, Acquisition of data, Drafting or revising the article, Contributed unpublished essential data or reagents; RBB, Conception and design, Drafting or revising the article, Contributed unpublished essential data or reagents; DBM, Conception and design, Analysis and interpretation of data, Drafting or revising the article; MY, Conception and design, Analysis and interpretation of data, Drafting or revising the article, Contributed unpublished essential data or reagents

## Author ORCIDs

Douglas B Menke, http://orcid.org/0000-0002-7109-1451
Michael D Shapiro, http://orcid.org/0000-0003-2900-4331

## Ethics

Animal experimentation: This study was performed in accordance with the recommendations in the Guide for the Care and Use of Laboratory Animals of the National Institutes of Health. All of the animals were handled and housed according to approved University of Utah institutional animal care and use committee (IACUC) protocols 10-05007 and 13-04012.

# Additional files

### Supplementary files

• Supplementary file 1. Primers used in this study. List of oligonucleotide names and targets for genotyping of indel on scaffold 79, qRT-PCR for gene expression assays, in situ hybridization probes, and pyrosequencing for allele specific expression assays.

### Major datasets

The following datasets were generated:

| Author(s) | Year | Dataset title | Dataset URL | Database, license, and accessibility information |
|---|---|---|---|---|
| Eric T Domyan, Zev Kronenberg, Carlos R Infante, Anna I Vickrey, Sydney A Stringham, Rebecca Bruders, Michael W Guernsey, Sungdae Park, Jason Payne, Robert B Beckstead, Gabrielle Kardon, Douglas B Menke, Mark Yandell, Michael D Shapiro | 2016 | Pomeranian pouter pigeon shotgun sequences | http://www.ncbi.nlm.nih.gov/bioproject/284526 | Publicly available at the NCBI BioProject (accession no. PRJNA284526) |
| Eric T Domyan, Zev Kronenberg, Carlos R Infante, Anna I Vickrey, Sydney A Stringham, Rebecca Bruders, Michael W Guernsey, Sungdae Park, Jason Payne, Robert B Beckstead, Gabrielle Kardon, Douglas B Menke, Mark Yandell, Michael D Shapiro | 2016 | Pigeon embryonic limb H3K27ac ChIP-seq data | http://www.ncbi.nlm.nih.gov/geo/query/acc.cgi?acc=GSE67875 | Publicly available at NCBI Gene Expression Omnibus (accession no. GSE67875) |

The following previously published dataset was used:

| Author(s) | Year | Dataset title | Dataset URL | Database, license, and accessibility information |
|---|---|---|---|---|
| Shapiro MD, Kronenberg Z, Li C, Domyan ET, Pan H, Campbell M, Tan H, Huff CD, Hu H, Vickrey AI, Nielsen SC, Stringham SA, Willerslev E, Gilbert MT, Yandell M, Zhang G, Wang J | 2013 | Whole genome sequencing of rock pigeon | http://www.ncbi.nlm.nih.gov/sra/?term=SRA054391 | Publicly available at the NCBI Sequence Read Archive (accession no. SRA054391) |

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
