## [Decision Letter]

Thank you for submitting your work entitled "A molecular shift in limb identity underlies the convergent evolution and development of feathered feet" for consideration by *eLife*. Your article has been reviewed by three peer reviewers, and the evaluation has been overseen by a Reviewing Editor, Marianne Bronner, and Detlef Weigel as the Senior Editor.

The reviewers have discussed the reviews with one another and the Reviewing editor has drafted this decision to help you prepare a revised submission.

This work represents an elegant use of pigeon genetics to address an important question of evolutionary morphology: the origin of feathered feet. A molecular understanding of this issue would open the door to advances in our knowledge of the scale to feather transition, the origin of feathered appendages, and other problems of significance. Using the tractable pigeon system, with a bevy of tools available for genetic and anatomical analysis, the authors reveal a correlation between feathered feet and a shift in expression of two regulatory factors, Pitx1 and Tbx5. The novelty is slightly dampened by the fact that the genes they found have been known to alter limb identify for more than a decade, but the work still represents a logical approach and valuable documentation, showing us how different approaches can lead scientists to the same important genes. While the genetic approach in pigeon is strong, the chicken variant part needs improvement as detailed below. Importantly, some of the discussion regarding the evolutionary implications seems over-interpreted and needs to be toned down.

1) A major problem that must be clarified and rephrased is the relationship to dinosaur feather/scale evolution. While this is certainly an interesting topic, the "four wing dinosaurs" found by Xing Xu and his associates clearly have leg skeleton in the hind feet. Thus, a "feathered limb" can have wing or leg skeleton inside. Consistent with this, Figure 4 nicely shows that the muffed pigeon foot exhibits some wing-like skeleton and muscle patterns. In dinosaur evolution, the change of feather and scale occurs in the dermis, and the skeleton remains foot-like. In contrast, the mutation studied here is a limb identity decision, which is a different matter and this must be clarified. It is an inappropriate analogy particularly to be in the first and last sentence of the manuscript.

2) "Convergent Evolution" used in the title is too strong and misleading. It should be rephrased in the title, Abstract and Introduction.

3) It is claimed that misexpression of a Pitx1-Enr construct disrupts scale formation and that this is consistent with phenotypes obtained following Pitx1 misexpression in the wing. What is shown in the two panels is a severely truncated and abnormal limb. It is not possible to assess any effect on scale formation from this part of the data. It seems to go too far to say that Pitx1-Enr is dominant-negative, when the luciferase assay merely indicates that the construct can repress transcription in a heterologous system. It is incorrect to state that “[…]verify that Pitx1EnR inhibited wild-type Pitx1 activity".

4) While the figure title states that "Tbx5 alters development of the foot epidermal appendages" the main text goes much further suggesting "misexpression of Tbx5[…] induced ectopic development of foot feathers". This is not at all clear from the panels in Figure 2—figure supplement 4B. Normal developing feathers are clear in other parts of the embryos at these stages and the structures highlighted do not look similar. The authors should be much more cautious about exactly what has occurred in these embryos.

5) An important issue is the localization of the cells that are expressing Tbx5 in the hindlimb, since it is epidermal derivatives that are being analysed largely and these structures are known to be patterned by signals from the mesoderm. It is stated that "ectopic Tbx5 expression was largely localized to the mesoderm of the proximal and posterior hindlimb" based on whole mount in situ hybridization. It would be clearer to show this in a section. It is further stated that this domain "shows a striking correlation with regions of epidermal transformation". This is not clear at all. Also, not too much can/should be read into the expression domain of Pitx1 shown for two breeds of pigeon. Are you certain that these are the correct stages to be analyzing the expression domains of these genes with respect to any possible action they may have on patterning epidermal structures? The same argument applies for the analysis of ectopic Tbx5.

7) The data in Figure 4 comparing two muscles and tibia/fibula in feral and Pomeranian pouter demonstrate a re-patterning of hind limb structures to more wing-like and that this can be attributed to the ectopic expression of Tbx5 described in different breeds (Indian fantail, English trumpeter). In regard to the fibula differences, it would certainly appear to be longer in the muffed pigeon but it remains a fibula in position and morphology. To suggest it is like an ulna is a misrepresentation. It looks nothing like an ulna in shape or articulation. The length of fibula can be different in different species and this need not be considered as a hindlimb to forelimb transformation.

Additional issues with this section are:

The section starts with a misrepresentation of the literature suggesting Tbx5 can result in muscular and skeletal mispatterning.

Any potential contribution of reduced Pitx1 expression to the morphological differences described is not considered.

Is "ulna" in subheading “Muffed pigeon breeds incur musculoskeletal patterning changes” a typo?

Is "lateral" hindlimb really intended. Is this really the primary site of ectopic Tbx5. This is described as 'posterior' in the earlier section.

8) In the Discussion, while it is mentioned that additional mutations also contribute to the muff phenotype it is assumed that these must all converge on the Tbx5 and Pitx1 pathways. This train of thought is continued in the suggestion that "a relatively small number of genetic changes are sufficient for a surprisingly large transformation of epidermal appendage morphology and distribution.” I don't believe the data make the case for 'sufficiency' in their argument since the misexpression data are not convincing and the birds could be harbouring a large number of additional mutations that contribute/are responsible for the feathering phenotype.

9) There is surprisingly little discussion of work on Tbx5 and Pitx1 function in the mouse. The authors must be aware that much of what they show and conclude is inconsistent with analyses of various mouse mutants of Tbx5 and Pitx1. This has been ignored largely, even though it would be exciting, if indeed true. The current interpretations of the results, without reference to the prior work in other systems, could further muddy an already messy literature on the roles of genes in determining the differences between fore limbs and hind limbs. It is therefore essential to clearly understand the functional significance of the QTL and expression analyses they have carried out. It would be interesting to test if elevating the levels of Pitx1 in a feathered breed would be sufficient to reduce feathers/replace with scales and reduce Tbx5 expression in the leg.

10) Have the authors considered whether the specific QTLs identified relate to expression of Tbx5 in the wing or other regions of the embryo or adult? What attempts have been made to demonstrate that the QTL are causative to the expression domain in the leg? Only one possible explanation (out of the many other alternatives) appear to have been explored and the final conclusion hangs on what appear to me to be unconvincing functional data.

11) Ptilopody refers to the presence of feathers in the shank. There are feathers on proximal parts of pigeon legs (both with scaled shank and feathered) that do not correlate with Tbx5 expression or reduction in Pitx1 expression. This is not mentioned or discussed. Is the assumption that the capacity of Tbx5 is localized to distal regions of the limb? There are of course feathers in many other regions of the bird where Tbx5 is not expressed also. Chicken breeds also show variation in feathering in the hindlimb-cochin and silkie chicken are analyzed to some extent. In silkie and some other chicken breeds (silky-feather) the basis of the mutation is known and/or causes other defects such as ploydactyly in Silkies. These do not correlate with the results and conclusions derived from this work in pigeon. These have not been discussed and are relevant to the significance of the proposed mechanisms in bird/dinosaur evolution.

12) The shared genotype for feathered feet between pigeon and chicken may be partial since some chicken with feathered feet have Tbx5 in the leg bud, but no difference in Pitx1, which is on chromosome 13 where the chicken ptilopody locus is located. Can the authors analyze different chicken ptilopody variants to qualify their convergent evolution statement?

[Editors' note: further revisions were requested prior to acceptance, as described below.]

Thank you for resubmitting your work entitled "Molecular shifts in limb identity underlie development of feathered feet in two domestic avian species" for further consideration at *eLife*. Your revised article has been favorably evaluated by Detlef Weigel (Senior editor), a Reviewing editor (Marianne Broner), and two reviewers. The manuscript has been improved but there are some remaining, relatively minor issues that need to be addressed before acceptance, as outlined below. We look forward to receiving a revised version that addresses these points:

Point 1: The point was to distinguish that there can be changes in feather and scale dermis patterns that are distinct from the other mesoderm derivatives (e.g. skeleton) rather than anything about dosage and/or stage requirements.

Point 7: Although the text has been changed, the authors still seem to be implying that the longer fibula is a partial hindlimb to forelimb transformation which it need not be (Minor issue:the ulna does not articulate at the wrist, the radius does.)

Please avoid the term “limb-specific”.

Subheading “Evolution of epidermal appendage distribution”: avialan to avian

---

## [Author Response]

This work represents an elegant use of pigeon genetics to address an important question of evolutionary morphology: the origin of feathered feet. A molecular understanding of this issue would open the door to advances in our knowledge of the scale to feather transition, the origin of feathered appendages, and other problems of significance. Using the tractable pigeon system, with a bevy of tools available for genetic and anatomical analysis, the authors reveal a correlation between feathered feet and a shift in expression of two regulatory factors, Pitx1 and Tbx5. The novelty is slightly dampened by the fact that the genes they found have been known to alter limb identify for more than a decade, but the work still represents a logical approach and valuable documentation, showing us how different approaches can lead scientists to the same important genes. While the genetic approach in pigeon is strong, the chicken variant part needs improvement as detailed below. Importantly, some of the discussion regarding the evolutionary implications seems over-interpreted and needs to be toned down.

We appreciate the positive feedback about our approach and results. We address our results in chicken directly in our responses to specific comments below. We also understand the concern about over-interpretation of the evolutionary implications of our results, and we have toned down our language, especially in the Abstract, Introduction, and Results. We still feel that our work provides exciting new insights about evolutionary changes in epidermal appendage identity and distribution, but we have made this less of a central theme of the paper.

1) A major problem that must be clarified and rephrased is the relationship to dinosaur feather / scale evolution. While this is certainly an interesting topic, the "four wing dinosaurs" found by Xing Xu and his associates clearly have leg skeleton in the hind feet. Thus, a "feathered limb" can have wing or leg skeleton inside. Consistent with this, Figure 4 nicely shows that the muffed pigeon foot exhibits some wing-like skeleton and muscle patterns. In dinosaur evolution, the change of feather and scale occurs in the dermis, and the skeleton remains foot-like. In contrast, the mutation studied here is a limb identity decision, which is a different matter and this must be clarified. It is an inappropriate analogy particularly to be in the first and last sentence of the manuscript.

We agree with this assessment, although we wish to point out a couple of items of potential confusion. First, the leg skeletons of muffed pigeons are clearly still legs, and we did not state otherwise in the original manuscript. To clarify this point, we now state in the Discussion:

“Collectively, these findings point to an alteration of the identity of the developing hind limb, rather than localized changes to individual epidermal placodes. […] Instead, different aspects of hind and forelimb morphology appear to have different dosage- and/or stage-dependent requirements.”

On this same theme, we do not claim that muscle mis-patterning in the hind limb directly resembles the normal forelimb. To emphasize this point, we added the following statement in the Discussion:

“These changes are aberrations of normal patterning, although they are not necessarily clear transformations to a more forelimb-like configuration.”

2) "Convergent Evolution" used in the title is too strong and misleading. It should be rephrased in the title, Abstract and Introduction.

This comment raises an interesting issue. Descent with modification under domestication is widely accepted as evolution, albeit not in a “natural” setting, but we realize that the language we used could lead to confusion. Our interpretation of convergence also seems to have no alternative – inheritance of feathered feet from the most recent common ancestor of pigeons and chickens is rejected by every recent phylogenetic hypothesis of avian evolution. We have altered our verbiage to the more descriptive “genetic mechanisms” and similar terms to avoid confusion and detract from the main messages of our study.

*3) It is claimed that misexpression of a Pitx1-Enr construct disrupts scale formation and that this is consistent with phenotypes obtained following Pitx1 misexpression in the wing. What is shown in the two panels (Figure 2—figure supplement 4B) is a severely truncated and abnormal limb. It is not possible to assess any effect on scale formation from this part of the data. It seems to go too far to say that Pitx1-Enr is dominant-negative, when the luciferase assay merely indicates that the construct can repress transcription in a heterologous system. It is incorrect to state that“*[*…]verify that Pitx1EnR inhibited wild-type Pitx1 activity".*

After further reflection, we agree that the direct implications of these functional experiments are difficult to assess. As a result, we chose to remove them from the revised manuscript.

*4) While the figure title states that "Tbx5 alters development of the foot epidermal appendages" the main text goes much further suggesting "misexpression of Tbx5[…] induced ectopic development of foot feathers". This is not at all clear from the panels in Figure 2*—*figure supplement 4B. Normal developing feathers are clear in other parts of the embryos at these stages and the structures highlighted do not look similar. The authors should be much more cautious about exactly what has occurred in these embryos.*

As mentioned above, we have removed these experiments from the manuscript.

*5) An important issue is the localization of the cells that are expressing Tbx5 in the hindlimb, since it is epidermal derivatives that are being analysed largely and these structures are known to be patterned by signals from the mesoderm. It is stated that "ectopic Tbx5 expression was largely localized to the mesoderm of the proximal and posterior hindlimb" based on whole mount in situ hybridization. It would be clearer to show this in a section.*

This is an excellent suggestion. We now include images of sections from each of the three pigeon phenotypes studied (scaled, small muff, large muff) in Figure 3—figure supplement 3. These sections helped us detect more dorsal expression than we were able to appreciate from whole embryos, and now describe “dorsal-posterior” expression of Tbx5 in the manuscript.

*It is further stated that this domain "shows a striking correlation with regions of epidermal transformation". This is not clear at all.*

Figure 3—figure supplement 3 now shows images from adult and embryonic muffed limbs. These images emphasize that the most pronounced epidermal transformations occur in the posterior (lateral digits in the adult) and dorsal aspects of the limb. The enlargement of the fibula (posterior/lateral zeugopod bone, Figure 5) also supports this interpretation.

Also, not too much can/should be read into the expression domain of Pitx1 shown for two breeds of pigeon. Are you certain that these are the correct stages to be analyzing the expression domains of these genes with respect to any possible action they may have on patterning epidermal structures? The same argument applies for the analysis of ectopic Tbx5.

We agree that the stages we chose to analyze might not represent differences (or similarities) that persist throughout development. To make this point clear, we have specified Hamburger-Hamilton stages in the revised manuscript. Further developmental biology work is needed to fully understand the specific time points at which the actions of these two genes are most important for specific aspects of epidermal, muscular, and skeletal development. However, our genetic and genomic results provide clear motivation for studying these two genes, and our allele-specific expression assays demonstrate that cis-regulatory differences specifically at these two loci affect their expression levels.

7) The data in Figure 4 comparing two muscles and tibia/fibula in feral and Pomeranian pouter demonstrate a re-patterning of hind limb structures to more wing-like and that this can be attributed to the ectopic expression of Tbx5 described in different breeds (Indian fantail, English trumpeter). In regard to the fibula differences, it would certainly appear to be longer in the muffed pigeon but it remains a fibula in position and morphology. To suggest it is like an ulna is a misrepresentation. It looks nothing like an ulna in shape or articulation. The length of fibula can be different in different species and this need not be considered as a hindlimb to forelimb transformation.

Thank you for highlighting these points of potential confusion. We agree that the fibula of muffed pigeons is still a fibula in both morphology and position, and we suspect that a typo (pointed out below) contributed to the impression that we were mistakenly identifying the fibula as an ulna. On the contrary, our intent was to convey that the fibula of the English trumpeter in Figure 4 (now Figure 5) is longer than usual and articulates with both the knee and the ankle, just as the ulna articulates with the elbow and wrist. We agree that fibular morphology varies widely among birds, but the configuration we observe in muffed birds is highly aberrant for a pigeon, and this is the key finding we tried to communicate. To emphasize this point, we modified the following sentence to refer specifically to pigeons instead of “birds” in general:

“We also found that the fibula, which is normally reduced in pigeons relative to the homologous forelimb structure (ulna), was enlarged[…]”

Further confusion might have originated from the typo in the following statement:

“Ectopic expression of Tbx5 in the hind limbs of chick embryos produces a similar enlargement of the ulna.”

In the above sentence, “ulna” should have been “fibula.” We have made this change in the revised manuscript.

Additional issues with this section are:

The section starts with a misrepresentation of the literature suggesting Tbx5 can result in muscular and skeletal mispatterning.

Thank you for this comment. Our effort to be economical with space unfortunately resulted in the following confusing statement:

“In mouse and chicken embryos, experimental manipulation of Pitx1 and Tbx5 expression can also result in muscular and skeletal mispatterning (Logan and Tabin 1999; Hasson, et al. 2010).”

We have now expanded the Discussion at the beginning of this section. We start with a similar overview statement, and provide additional commentary to de-convolute the known roles of these two genes on muscular and skeletal patterning in different vertebrates.

“In mouse and chicken embryos, experimental manipulation of Pitx1 and Tbx5 expression can result in muscular and skeletal abnormalities. […] Given the dramatic musculoskeletal defects observed in other organisms with experimentally altered Pitx1 and Tbx5 expression, we compared the hind limb morphology of adult feral pigeons[…]”

Any potential contribution of reduced Pitx1 expression to the morphological differences described is not considered.

As mentioned above, and as detailed in the Discussion (“Roles of Pitx1 and Tbx5 in diversity and disease” subheading), we now discuss the impact of Pitx1 misexpression and mutant alleles in a bird, two mammals, and a fish. These examples illustrate that decreases in Pitx1 expression can result in skeletal changes, although we are not presently able to determine the specific effects of Pitx1 versus Tbx5 on musculoskeletal morphology in muffed pigeons. Follow-up work will entail detailed phenotyping of the musculoskeletal system of our F2 cross by dissection and CT analysis, and these phenotypes will allow us to separate the effects of these two loci on musculoskeletal changes. To emphasize this point, we now state in the aforementioned subsection of the Discussion:

“Our ongoing analyses of musculoskeletal phenotypes in our F2 cross, which includes individuals with different combinations of feathered-foot alleles of Pitx1 and Tbx5, will allow us to understand the separate and epistatic effects of these loci on musculoskeletal anatomy.”

Is "ulna" in subheading “Muffed pigeon breeds incur musculoskeletal patterning changes”a typo?

Yes, thank you for identifying this error. “Ulna” has been changed to “fibula” in the revised manuscript.

Is "lateral" hindlimb really intended. Is this really the primary site of ectopic Tbx5. This is described as 'posterior' in the earlier section.

Thank you for pointing out this discrepancy. The posterior limb bud of the embryo becomes the lateral side of the adult limb. To clarify this point, we have altered the original statement to read:

“Notably, all of the modified structures of ptilopodous pigeons develop in the posterior (lateral in the adult) and dorsal hind limb, which are the primary sites of ectopic Tbx5 expression.”

8) In the Discussion, while it is mentioned that additional mutations also contribute to the muff phenotype it is assumed that these must all converge on the Tbx5 and Pitx1 pathways. This train of thought is continued in the suggestion that "a relatively small number of genetic changes are sufficient for a surprisingly large transformation of epidermal appendage morphology and distribution.” I don't believe the data make the case for 'sufficiency' in their argument since the misexpression data are not convincing and the birds could be harbouring a large number of additional mutations that contribute/are responsible for the feathering phenotype.

We agree that other genes are probably involved in the foot feathering phenotype. However, multiple lines of evidence demonstrate that only two loci have major effects on the phenotype. First, classical genetic studies implicate major effects of the grouse and slipper loci, the molecular identities of which were previously unknown. Second, our QTL mapping study implicated just two loci of major effect on the phenotype. Importantly, our F2 intercross led to the recombining of muffed and scaled genomes, thereby minimizing the effects of the parental genetic backgrounds on phenotypic variation in the F2 offspring. If other loci had significant effects on the phenotype, we would have observed additional significant QTL. As the size of our F2 population increases, we might have enough power to map additional loci that make small contributions to the phenotype. Some of these loci might converge directly on the Tbx5 and Pitx1 pathways, but we see no reason to assume that they will. However, this does not diminish the result that the two QTL we mapped in this study do indeed control a large proportion of phenotypic variance in the cross. Third, our genome scans comparing scale-footed and feather-footed breeds independently detected the same two chromosome regions as our QTL mapping study. If other loci exerted major effects on the phenotype or were selected repeatedly across breeds, we would have detected them in our selection scans. It remains possible that other loci have minor effects on the phenotype, and we might not have the power to detect them in our whole-genome scans of scale-footed and feather-footed breeds. Our results also do not exclude the possibility of allelic heterogeneity at modifier loci, which would make them harder to detect. However, these possibilities do not diminish the two highly significant associations we detected with our current data set. We agree that our misexpression data do not make a clear case for developmental sufficiency, but our results robustly support genetic sufficiency of these loci for foot feathering.

9) There is surprisingly little discussion of work on Tbx5 and Pitx1 function in the mouse. The authors must be aware that much of what they show and conclude is inconsistent with analyses of various mouse mutants of Tbx5 and Pitx1. This has been ignored largely, even though it would be exciting, if indeed true. The current interpretations of the results, without reference to the prior work in other systems, could further muddy an already messy literature on the roles of genes in determining the differences between fore limbs and hind limbs. It is therefore essential to clearly understand the functional significance of the QTL and expression analyses they have carried out. It would be interesting to test if elevating the levels of Pitx1 in a feathered breed would be sufficient to reduce feathers/replace with scales and reduce Tbx5 expression in the leg.

We agree that this is an important topic in the evolution and development of fore- and hind limb identity. In our response to comment 7, we provide a more in-depth discussion of previous work in both the mouse and chicken experimental systems. In short, we find that regulatory alterations to Tbx5 in pigeon and chicken breeds align nicely with previous functional data in chickens. We believe this work doesn't muddy the waters, but rather bolsters confidence in the original Tbx5 overexpression experiments in chickens. Pitx1 and Tbx5 undoubtedly retain similar functions among tetrapods (and other vertebrates), but we find no reason to presume that gene regulation will work identically in avians and mammals. Moreover, the common ancestor of mammals and birds lived more than 300 million years ago and the ectodermal structures that are the focus of this study (feathers) are not even present in mammals. Horton et al. (2008, Dev Genes Evol) speculate about an FGF-mediated mechanism that could explain part of the difference in the conflicting data for the role of Tbx5 in forelimb identity, and we now cite this paper in the manuscript. For example, in the Discussion, we now state:

“We note that, although genetic manipulations indicate that Tbx5 does not specify forelimb identity in mice, the divergence time between mammals and birds is deep (>300 million years) and subtly different roles for this transcription factor in limb outgrowth and identity might have evolved in these lineages (Minguillon et al. 2005; Horton et al. 2008).”

We also agree that the suggested Pitx1 misexpression experiment is worthwhile. We hope to conduct this experiment soon.

10) Have the authors considered whether the specific QTLs identified relate to expression of Tbx5 in the wing or other regions of the embryo or adult? What attempts have been made to demonstrate that the QTL are causative to the expression domain in the leg? Only one possible explanation (out of the many other alternatives) appear to have been explored and the final conclusion hangs on what appear to me to be unconvincing functional data.

Thank you for pointing out these important issues that need clarification. It is indeed possible that expression of Tbx5 is altered in other regions of the embryo; however, the embryonic wing is not one of these regions, as demonstrated by the expression assays reported in Figure 3—figure supplement 1 (Tbx5 forelimb expression among breeds is statistically indistinguishable). Moreover, whole-mount in situ hybridization of Tbx5 only revealed major Tbx5 expression pattern changes in the hind limbs of muffed embryos. Crucially, the two significant QTL and genomic regions were identified specifically and exclusively based on hind limb feathering phenotypes. These genomic regions might have an effect on development and feathering of other structures, but this issue was not addressed in, and is largely irrelevant to, our study. Two separate methodologies – QTL mapping and whole-genome scans – converged on the same two loci, and only these two loci, that exert quantitative (QTL mapping) and qualitative (genome scans) effects on foot feathering. After identifying these two loci based specifically on hind limb phenotypes, we expected to see coding changes – we did not find any fixed differences – or limb bud expression differences among breeds with different hind limb phenotypes. Therefore, we assayed limb bud expression among breeds to check the viability of strong candidate genes in these regions, and then showed definitively that Pitx1 and Tbx5 have cis-regulatory changes in a hybrid background. The hybrid background is important because alleles inherited from each parent are responding to the same trans-acting factors in a common cellular environment; thus, any differences in expression are attributable to cis-regulatory differences between parental alleles. This allele-specific expression experiment is crucial because it shows that expression differences in Pitx1 and Tbx5 are not just different among phenotypes, but they are also heritable: the feathered-foot allele of Tbx5 is expressed at a higher level in the hybrid hind limb bud, and the feathered-foot allele of Pitx1 is expressed at a lower level. In contrast, similar expression levels of different alleles in the hybrid background would confirm that differences among phenotypes are solely attributable to trans-acting regulatory changes (e.g., differential expression of an upstream transcription factor); however, we did not observe this outcome for either Pitx1 or Tbx5. To emphasize the importance of this experiment, we have moved the explanation of this experiment and its results into a new figure in the main text (Figure 4, formerly part of Figure 3—figure supplement 1). Further supporting our interpretation, the genome-wide H3K27ac ChIP-seq enrichment differences in limb buds from breeds with different limb phenotypes were significant in regions near Pitx1 and Tbx5. All of these analyses were specific to the embryonic (gene expression) or adult (phenotypes) hind limbs. Future experiments could indeed explore the role of Tbx5 in generating variation in feathering or other traits elsewhere in the embryo and adult; however, we intentionally drew genomic data from a wide variety of feather-footed breeds that do not share other obvious traits in common. Therefore, we would not expect to obtain similar results for other traits with our current data set.

*11) Ptilopody refers to the presence of feathers in the shank. There are feathers on proximal parts of pigeon legs (both with scaled shank and feathered) that do not correlate with Tbx5 expression or reduction in Pitx1 expression. This is not mentioned or discussed. Is the assumption that the capacity of Tbx5 is localized to distal regions of the limb? There are of course feathers in many other regions of the bird where Tbx5 is not expressed also.*

We believe that the presence of feathers throughout the body of a pigeon (including the regions shown in Figure 2, in which Tbx5 is not expressed) is well known, and is also displayed in the phenotypes of interest in Figure 1. The presence of feathers in other parts of the limb are now also shown in Figure 3—figure supplement 3. Our focus for this paper is on sites of variation in scaled and feathered hind limb epidermis, which include only the distal portion of the limb. We did not investigate the roles of Tbx5 and Pitx1 in feather variation in other parts of the body (please also see our response to the previous comment).

Chicken breeds also show variation in feathering in the hindlimb-cochin and silkie chicken are analyzed to some extent. In silkie and some other chicken breeds (silky-feather) the basis of the mutation is known and/or causes other defects such as ploydactyly in Silkies. These do not correlate with the results and conclusions derived from this work in pigeon. These have not been discussed and are relevant to the significance of the proposed mechanisms in bird/dinosaur evolution.

We agree that our results on foot feathering in pigeons conflict somewhat with previous results in chicken models. However, to our knowledge, the specific mutation(s) leading to foot feathering in silkies and other breeds are not known. Indeed, we believe that the identification of specific candidate genes contributes to the novelty of our work in pigeons. A previous genetic mapping study implicated a chromosomal region around Pitx1 in ptilopody in silkies, but specific genes and mutations were not identified. In fact, the Pitx1 gene was not even mentioned as a candidate gene in that study (Dorshorst et al., 2010, J Hered). Polydactyly in silkies is a different genetic issue than ptilopody (these two loci are unlinked), and we do not address this issue in the manuscript. Ptilopodous silkies can be polydactyl, but not all are. Silky feathering throughout the body is also unlinked to ptilopody. We agree that the disparity between chickens and pigeons is important and we address it in some detail in the last paragraph of the Results section:

“Classical genetic studies implicate at least two loci in heavy foot feathering in chickens (Punnett and Bailey 1918; Lambert and Knox 1929; Warren 1948; Somes 1992), although the molecular genetic origins of the trait remain unknown. […] Furthermore, different populations of breeds such as silkies appear to have different constellations of ptilopody loci and alleles, and it is possible that we used strains that do not have Pitx1 mutations (Wexelsen 1934; Somes 1992).”

12) The shared genotype for feathered feet between pigeon and chicken may be partial since some chicken with feathered feet have Tbx5 in the leg bud, but no difference in Pitx1, which is on chromosome 13 where the chicken ptilopody locus is located. Can the authors analyze different chicken ptilopody variants to qualify their convergent evolution statement?

The common ancestor of chickens and pigeons did not have feathered feet, so we believe that the feathered feet of these two species is best described as a case of phenotypic convergence. If the reviewer is referring to breed “variants,” we already examined gene expression in two ptilopodous breeds, the Cochin and silke. None of our gene expression data point to a role for Pitx1 in ptilopody in silkies or Cochins, but as we state in the manuscript, we could be examining the wrong stage to implicate this gene. However, we demonstrate differences in Tbx5 expression between the hind limb buds of wild-type and ptilopodous breeds. We also generated silkie x white leghorn hybrid embryos and showed that ectopic expression of the feathered-foot Tbx5 allele is due (at least in part) to cis-regulatory changes, but we do not have genetic mapping or enough genomic resequencing data from chickens to directly test for involvement of specific sequence variants.

To emphasize that our results might account for some, but not all, of the ptilopody phenotype in chickens, we have added “in part” to the following statement:

“Hence, Tbx5-related developmental mechanisms may, in part, underlie the evolution of foot feathering in two species that last shared a common ancestor over 80 million years ago.”

[Editors' note: further revisions were requested prior to acceptance, as described below.]

Point 1: The point was to distinguish that there can be changes in feather and scale dermis patterns that are distinct from the other mesoderm derivatives (e.g. skeleton) rather than anything about dosage and/or stage requirements.

Thank you for this clarification. We have addressed this comment in the revised manuscript, and the paragraph in question now reads:

“Collectively, these findings point to a partial alteration of the identity of the developing hindlimb, rather than localized changes to individual epidermal placodes. […] Therefore, we propose that different aspects of fore- and hindlimb morphology could have different dosage- and/or stage-dependent requirements for exposure to identity cues.”

We chose to retain a modified version of the last sentence in the revised manuscript because we believe it proposes an important and testable research hypothesis for future work.

Point 7: Although the text has been changed, the authors still seem to be implying that the longer fibula is a partial hindlimb to forelimb transformation which it need not be (Minor issue:the ulna does not articulate at the wrist, the radius does.)

We have altered this passage to highlight a previous interpretation about the enlargement of the fibula in experimentally induced hindlimb expression of Tbx5. We agree that the longer fibula of pigeons need not necessarily be due to a partial hindlimb to forelimb transformation, but given the strong evidence that epidermal identity is transformed, we believe we are justified in raising this issue and would be downplaying the significance of important experimental interpretations from others’ work by not doing so. We realize that not everyone will agree with these interpretations, but we feel that it is our responsibility to credit these experimental and interpretive precedents. The revised passage reads as follows:

“We also found that the fibula, which is normally splint-like and shorter than the tibia in pigeons, was enlarged (Figure 5) and two phalanges of digit 4 were fused in feather-footed breeds (not shown). […] However, experimental ectopic expression of Tbx5 in the hindlimbs of chick embryos produces an enlargement of the fibula reminiscent of extreme pigeon phenotypes, and Takeuchi et al. (1999) compared this morphology to a forelimb-like condition (the fibula “makes a joint at its distal end like a normal ulna [the corresponding postaxial zeugopod bone of the forelimb].”

To tone down our argument further, we replaced “transformation” with “changes” in the following concluding sentence:

“Thus, the morphological changes to the hindlimbs of feather-footed pigeon breeds are considerably more than skin deep.”

We also deleted from the caption for Figure 5 a statement that mentions a comparison between the fibula and ulna.

Regarding the minor issue, we cannot find any description in the comparative anatomy literature that supports the statement about a lack of articulation between the ulna and wrist in avians, nor does our own experience with skeletal preparations of embryonic or adult pigeon material support this statement.

Please avoid the term limb-specific

Without guidance about the reason to modify this term, we are somewhat confused about how best to address this request. “Limb-specific” (including “forelimb-specific” and “hindlimb-specific”) is commonly used throughout the limb development and morphology literature to describe gene expression patterns that are specific to one or another set of limbs. Many of the instances of “limb-specific” in the previous version of the manuscript resulted from an earlier request to change “hindlimb” (the standard British spelling and a widely accepted format in the anatomical and developmental biology literature) to “hind limb”; therefore, “hindlimb-specific” (a term in widespread use) became “hind limb-specific,” which is comparatively unusual and awkward. We note that even the first and second round of reviewer comments on this manuscript use “hindlimb” in every case (e.g., Point 7, above), with the exception of the comment in the first review to make the change from “hindlimb” to “hind limb.” Therefore, with clarity and convention in mind, we elected to change “hind limb” to “hindlimb” in the revised manuscript.

To further clarify what is meant by limb specificity, we have expanded the descriptions of Pitx1 and Tbx5 limb expression domains when these genes are first mentioned:

“The highest pFst peak on scaffold 79 – corresponding to the major-effect QTL on LG 11 for the proportion of tarsometatarsal feathering – was approximately 200 kb upstream of Pitx1, a gene that encodes a homeobox-containing transcription factor that is normally expressed in the vertebrate hindlimb but not the forelimb (Figure 2). The highest pFst peak on scaffold 70 – corresponding to the major-effect QTL on LG 20 for toe feather length – was 40 kb upstream of Tbx5, a gene that encodes a T-box transcription factor that is normally expressed in the vertebrate forelimb but not the hindlimb (Figure 2).”

We made several other changes to cut down on the number of times “fore/hindlimb- specific” appears in the text, but we believe this terminology is usually the most meaningful, accurate, and clear way to convey information. We welcome suggestions for alternative terminology.

Subheading “Evolution of epidermal appendage distribution” avialan to avian

This change would alter the meaning of the sentence, so we elected to delete the word entirely. Doing so actually broadens the scope of the question we are asking, which we see as a positive outcome:

“How might pigeons help us understand the evolution of epidermal appendage distribution and limb morphology in other species?”

We also use “avialan” later in the same paragraph to convey an interpretation by Zheng et al. 2013. Changing this term to “avian” would be incorrect and change the meaning of the statement.